# Learning to Rank Features to Enhance Graph Neural Networks for Graph Classification

**Fouad Alkhoury**                                            *alkhoury@iai.uni-bonn.de*
*University of Bonn*
*Lamarr Institute for Machine Learning and Artificial Intelligence*

**Tamás Horváth**                                            *horvath@cs.uni-bonn.de*
*University of Bonn*
*Fraunhofer IAIS*
*Lamarr Institute for Machine Learning and Artificial Intelligence*

**Christian Bauckhage**                                            *bauckhage@iai.uni-bonn.de*
*University of Bonn*
*Fraunhofer IAIS*
*Lamarr Institute for Machine Learning and Artificial Intelligence*

**Stefan Wrobel**                                            *wrobel@cs.uni-bonn.de*
*University of Bonn*
*Fraunhofer IAIS*
*Lamarr Institute for Machine Learning and Artificial Intelligence*

**Reviewed on OpenReview:** *https://openreview.net/forum?id=WmZGuWRAWb*

## Abstract

A common strategy to enhance the predictive performance of graph neural networks (GNNs) for graph classification is to extend input graphs with node- and graph-level features. However, identifying the optimal feature set for a specific learning task remains a significant challenge, often requiring domain-specific expertise. To address this, we propose a general two-step method that automatically selects a compact, informative subset from a large pool of candidate features to improve classification accuracy. In the first step, a GNN is trained to estimate the importance of each feature for a given graph. In the second step, the model generates feature rankings for the training graphs, which are then aggregated into a global ranking. A top-ranked subset is selected from this global ranking and used to train a downstream graph classification GNN. Experiments on real-world and synthetic datasets show that our method outperforms various baselines, including models using all candidate features, and achieves state-of-the-art results on several benchmarks.

## 1 Introduction

Graph neural networks (GNNs) (see, e.g., Hamilton, 2020) are widely used for learning from graph-structured data, especially in node and graph classification tasks. However, standard GNNs are limited by the expressive power of the 1-dimensional Weisfeiler-Leman test (Morris et al., 2019), which restricts their ability to distinguish certain non-isomorphic graphs. This limitation is particularly problematic in practical learning scenarios where structural differences are also task-relevant. A common strategy to address this issue is to enrich GNNs with additional feature information. This is especially crucial for graph classification, the focus of this work, where effective graph representations depend on both node- and graph-level features. Yet, the success of this approach strongly depends on the relevance and quality of the features provided.

Merely supplying GNNs with a large set of task-independent features does not guarantee improved performance. Since GNNs treat all input features as potentially relevant, their implicit feature extraction mechanisms cannot fully eliminate the negative influence of irrelevant or redundant features on generalization. In particular, such features may dominate and distort message passing, leading to degraded quality of learned representations. This issue becomes especially critical in graph classification, where global representations aggregate information from many nodes, amplifying the negative effect of noisy or irrelevant features. For example, in molecular property prediction, only a subset of atomic descriptors typically influences activity (Ponzoni et al., 2019).

Existing methods for incorporating additional features typically rely on predefined static feature sets (Barceló et al., 2021; Bouritsas et al., 2023; Cui et al., 2022; Duong et al., 2019), which require substantial domain expertise and often fail to generalize across datasets. More recent work on dynamic feature selection for GNN node classification (Naik et al., 2024) offers more flexibility but focuses on local graph patterns, neglecting global features crucial for graph-level tasks. In a recent work (Alkhoury et al., 2025), we proposed a feature-ranking GNN that learns to select relevant features for node classification on unseen graphs, improving accuracy and reducing computational cost. However, it is limited to node classification and cannot be directly extended to graph classification, where global feature selection is essential.

These gaps reveal that while GNNs possess implicit feature extraction capabilities, they remain sensitive to noisy and irrelevant features and lack principled mechanisms for targeted, data-driven feature selection at the graph level. To overcome these limitations, we propose a general framework that automates feature selection through a dedicated GNN architecture, jointly identifying informative local and global features during representation learning. Our approach enhances graph classification performance beyond what is possible via implicit feature extraction alone, offering a new perspective on how explicit feature selection in GNNs can improve predictive accuracy while reducing training and inference time.

Building on the framework we introduced in Alkhoury et al. (2025), we present an automatic and scalable feature selection method for *graph classification*. Our method proceeds in two major steps. First, given a set of training graphs, we sample a small, random subset of the data and compute all features from a universal pool for each graph in this subset, resulting in feature vectors of uniform dimensionality. These vectors, along with their corresponding class labels, are then used for feature ranking. To generate these rankings, we considered three different approaches: random forests, support vector machines, and local explanation methods from explainable AI. Among them, random forests consistently delivered the most reliable feature rankings. Following the approach we proposed in Alkhoury et al. (2025), we then train a feature-ranking GNN (FR-GNN) using the original graphs and their feature rankings. This enables the FR-GNN to predict feature importance for all remaining graphs in the training set without the need to compute any features. The local rankings are aggregated into a single global ranking, which is then applied to all training graphs in the second step and to unseen graphs later during class prediction.

In the second step, the top $K$ features from the global ranking are selected and computed for all training graphs. A graph classification GNN (GC-GNN) is then trained using these graphs, augmented with only the $K$ selected features. Since both feature selection and ranking rely on a small subset of data and a limited number of features, the overall process remains computationally feasible.

To assess the effectiveness of our approach, we conducted experiments on both real-world and synthetic datasets, utilizing 124 local and global graph features as well as three GNN architectures (GCN (Kipf & Welling, 2016), GAT (Velickovic et al., 2018), and GraphSAGE (Hamilton et al., 2017)), and compared our method to three baseline approaches.

On the real-world datasets, using only the top six features selected by our method ($K = 6$), the trained GC-GNN consistently outperformed the baseline that used no features, demonstrating the importance of incorporating features. Furthermore, our method surpassed the baseline that used all 124 features by an average margin of 7%, suggesting that including irrelevant or redundant features can lead to overfitting and degrade performance. Additionally, it consistently outperformed GC-GNNs trained on six randomly selected features, indicating that the features chosen by our method are genuinely informative rather than selected by chance. Notably, for the real-world datasets where state-of-the-art (SOTA) algorithm results are

publicly available (together with code), our method either outperforms or achieves comparable predictive performance.

To further evaluate robustness, we generated synthetic graphs using various random graph models with increasing classification difficulty. This was achieved by attaching graph motifs (graphlets) and defining target classes through logical formulas of varying complexity, where literals indicate the presence or absence of specific motifs. Our method consistently outperformed the baselines that used no features or random features, even on the most challenging tasks. While the baseline using all features achieved high accuracy ($\geq 95\%$) on synthetic data, our method using just six selected features often performed slightly better. In one particularly complex case involving four classes defined by nested logical conditions, the all-feature baseline outperformed our approach; however, increasing our selection to the top 20 features ($K = 20$) narrowed the performance gap. This result confirms that our method remains competitive even as task complexity increases.

The rest of the paper is organized as follows. Section 2 overviews the related work. Section 3 describes the proposed method. The experimental results are reported in Section 4. Finally, in Section 5 we conclude and mention some problems for future work.

## 2 Related Work

Research on graph classification spans a wide spectrum of approaches, from substructure-based kernels to deep neural architectures. A central challenge across these methods is the identification of informative features while minimizing redundancy and noise. Since our framework focuses on effective feature ranking and selection, we concentrate on methods most relevant to this aspect of graph classification. We organize related work into five categories and discuss how our approach differs from each.

**Substructure and Graphlet-Based Methods** A common approach to graph classification relies on *graph kernels*, such as the *graphlet kernel* (Shervashidze et al., 2009), which measures graph similarity by counting small subgraphs (graphlets). While polynomial-time enumeration is feasible for small graphs, it becomes computationally prohibitive for large ones. To address this limitation, efficient subgraph estimation techniques have been proposed, such as *graphlet counts approximation* without full enumeration (Rossi et al., 2018). More recently, the Substructure Assembling Network (SAN) (Yang et al., 2022) learns graph representations by hierarchically composing local substructures with a recurrent unit and soft attention mechanism, enabling the construction of discriminative graph-level features. Our method differs from these approaches in two key aspects. First, rather than exhaustively enumerating or estimating substructures, we rank and select a compact subset of candidate features *without* computing them across the entire training data, substantially reducing overhead. Second, our candidate feature set extends beyond substructures to include global graph properties and statistical summaries of node features, providing a richer and more versatile feature space for graph classification.

**Feature Selection and Noise Reduction** Real-world graphs often exhibit noise, sparsity, and redundant connections (Dai et al., 2022), making it difficult to extract meaningful patterns. Several approaches have been proposed to to mitigate the effects of noise or irrelevant information. For example, Fu et al. (2020); Ma et al. (2021) present theoretical frameworks interpreting GNN aggregation as a graph signal denoising process, suggesting that GNNs smooth node features to reduce noise. ES-GNN (Guo et al., 2024) explicitly distinguishes relevant from irrelevant edges to prevent the propagation of uninformative connections. Unlike these works, our method filters and ranks features prior to GNN training, achieving both computational efficiency and improved predictive accuracy. In fact, our experiments show that using the full feature set degrades performance, underscoring that GNNs alone cannot fully suppress the influence of irrelevant or redundant features.

**Dual-Network Architectures** Several works employ dual-network architectures to jointly address feature selection and classification. For example, Akyol et al. (2021) use two GNNs within a variational autoencoder for action recognition and prediction, while Maurya et al. (2023) propose a Dual-Net GNN for node classification, where one network selects node features for the other. Similarly, DualNetGO (Chen & Luo, 2024) alternates between selector and classifier networks for protein function prediction, starting from randomly

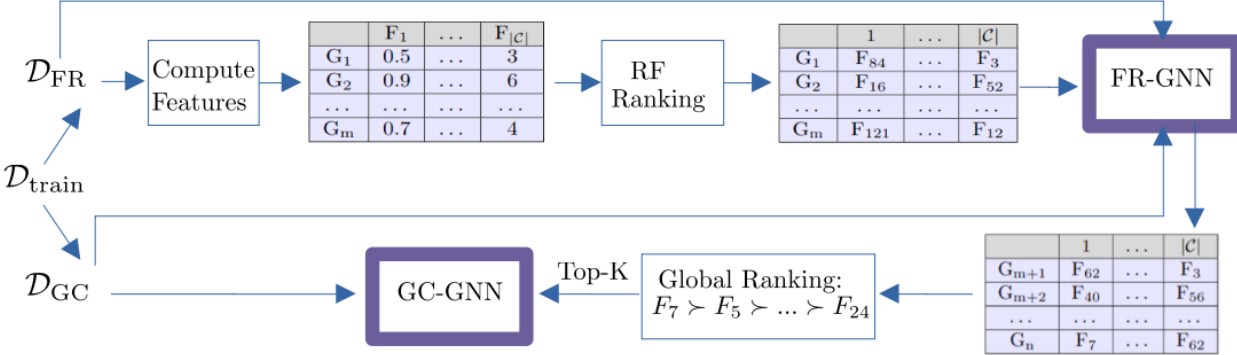

Figure 1: High-level overview of the FR-GNN training step (top) and the GC-GNN training step (bottom).

combined features. Our approach also adopts two GNNs, but with key differences: (i) it operates at the graph-level rather than for node classification, (ii) the feature ranking and classification stages are explicitly decoupled, simplifying optimization, and (iii) instead of relying on random initialization or domain-specific heuristics, it integrates a feature ranking stage to begin with a compact and relevant feature set.

**Graph Representation Learning and Pooling** Other approaches learn graph-level representations without explicit feature selection. Fei & Huan (2008) identify frequent subgraphs based on spatial distribution consistency, whereas we use subgraph counts as explicit features. HGP-SL (Zhang et al., 2019) combines hierarchical graph pooling with structure learning, implicitly refining the graph topology via learned node importance scores. In contrast, our method follows a feature-driven strategy: we rank and select informative graph-level features prior to training using an interpretable model, namely random forests. Cui et al. (2022) investigate the impact of hand-crafted features on GNN performance but without ranking or selecting features, unlike our approach.

**GNN-Based Feature Ranking Learning** In our closely related recent work (Alkhoury et al., 2025), we proposed a framework to improve predictive performance in node classification by identifying important features using a random forest-based selection mechanism. In contrast, our approach focuses on graph classification and explores both node-level and graph-level features, requiring entirely different techniques. In particular, while the feature-ranking GNN classifier in (Alkhoury et al., 2025) is trained on synthetic datasets designed to exhibit structural diversity, our method operates directly on the training graphs of the target dataset, ensuring that the resulting feature ranking reflects the specific patterns relevant to the classification task at hand. Another key difference lies in the aggregation strategy: whereas their method produces a separate feature ranking for each input graph (as needed for node classification), our approach aggregates feature importance scores across all training graphs, resulting in a dataset-level feature ranking suitable for graph classification.

## 3   The Method

This section presents our method[1] for automatically selecting a compact subset of features from a large pool of candidates, with the goal of improving the predictive performance of a GC-GNN using the selected additional features. Our method consists of two steps (see Fig. 1 for a schematic overview of the process).

**(Step 1) Training a Feature Ranking GNN (FR-GNN):** Given a set of training graphs $\mathcal{D}_{\text{train}}$, we first select a small random set $\mathcal{D}_{\text{FR}} \subseteq \mathcal{D}_{\text{train}}$, compute a feature ranking for each graph in $\mathcal{D}_{\text{FR}}$ using random forests, and train an FR-GNN on these graphs and their corresponding feature rankings. This enables the FR-GNN to learn to predict feature importance rankings for unseen graphs.

---

[1]Code is available at github.com/FouadAlkhoury/GraphClassificationRankingFeatures.

Table 1: Overview of the 124 features used in the experiments, grouped into three categories: graphlets, aggregated node features, and global graph properties.

| Category | Description |
| --- | --- |
| Graphlet Features (62) | 31 count features and 31 binary indicator features for all connected, non-isomorphic subgraphs (graphlets) of 2-5 nodes, plus the hexagon |
| Aggregated Node Features (56) | Statistics (mean, std, skewness, kurtosis) for degree, centrality measures (eigenvector, closeness, harmonic, and betweenness), largest first coloring, number of edges within egonet, node clique number, number of cliques, clustering coefficient, square clustering coefficient, pagerank, hubs, core number |
| Global Graph Properties (6) | number of nodes, number of edges, diameter, avg. and std. of path length, random feature |

**(Step 2) Training a Graph Classification GNN (GC-GNN):** We then take the remaining training data $\mathcal{D}_{\mathrm{GC}} = \mathcal{D}_{\mathrm{train}} \setminus \mathcal{D}_{\mathrm{FR}}$, use the trained FR-GNN to predict a feature ranking for each graph in $\mathcal{D}_{\mathrm{GC}}$, and aggregate these individual rankings into a global ranking. The top $K$ features from this global ranking are then selected, computed for all graphs in $\mathcal{D}_{\mathrm{GC}}$, and used to augment them with corresponding feature vectors of length $K$. A GC-GNN is subsequently trained on these augmented graphs to classify unseen graphs using only the most informative features.

We discuss these two steps in detail in Sections 3.2 and 3.3 below. Before that, in Section 3.1, we first discuss the types of candidate graph-level features used in our experimental evaluation.

## 3.1 Candidate Features

The success of our approach depends, among other factors, on the appropriate selection of a *universal* pool of candidate features. This pool is not tailored to a specific task. Instead, it should capture a broad range of global graph statistics and structural properties, which can either be intrinsic to the graph or derived by aggregating node-level features. Capturing such structural properties is crucial for the quality of the trained model, especially when graphs from the same target class are expected to share common patterns that distinguish them from graphs of other classes. Aggregating node-level features into graph-level representations allows us to embed graphs into a common feature space, thereby facilitating effective graph classification.

**Graphlet Features** These features are computed directly over the entire graph and are used in both binary and counting forms. They encode the presence (binary form) or frequency (counting form) of small subgraphs, known as *graphlets*, such as, for example, triangles or 4-cliques. The counts of these graphlets capture how often characteristic structural patterns appear as subgraphs, providing highly indicative information about the graph's properties and potentially capturing structural differences between graphs from different classes.

**Aggregated Node Features** In contrast to graphlets, aggregated node features (e.g., node degree, betweenness centrality) are computed at the node level before being summarized to form graph-level features. To derive graph-level features from a particular node feature for a graph $G$, we compute statistical summaries characterizing the feature's distribution across all nodes of $G$. Specifically, we use the following four descriptive statistics for aggregation: (1) *mean*, (2) *standard deviation*, (3) *skewness*, defined by $\frac{n}{(n-1)(n-2)} \sum_{i=1}^{n} \left(\frac{x_i - \bar{x}}{s}\right)^3$, measuring the asymmetry of the distribution, and (4) *kurtosis*, defined as $\frac{1}{n} \sum_{i=1}^{n} \left(\frac{x_i - \bar{x}}{s}\right)^4 - 3$, quantifying the peakedness or flatness of the distribution relative to the normal distribution. Here, $n$ denotes the number of nodes in $G$, $x_i$ the node feature value for node $i$, $\bar{x}$ the mean of the node feature values, and $s$ their standard deviation.

---

**Algorithm 1** Learning Feature Ranking GNN

---

**Input:** set $\mathcal{D}_{\text{FR}}$ of labeled graphs, candidate feature set $\mathcal{C}$
**Output:** FR-GNN model $\Phi_{\text{FR}}$

---

1:  **for all** $G \in \mathcal{D}_{\text{FR}}$ **do**
2:      compute the feature vector $\vec{x}_G$ defined by $\mathcal{C}$ for $G$
3:  $\mathcal{D} \leftarrow \emptyset$
4:  **for all** $G \in \mathcal{D}_{\text{FR}}$ **do**
5:      $\mathcal{G}_G = \{G' : G' \in \mathcal{D}_{\text{FR}} \text{ and } \text{CLASS}(G') \neq \text{CLASS}(G)\}$
6:      **for** $i = 1, \ldots, t$ **do**
7:          select a number $k$ uniformly at random from $\{3, 4, 5\}$
8:          select a random subset $\mathcal{G}'$ of $\mathcal{G}_G$ with $|\mathcal{G}'| = k$
9:          train a random forest $F_i$ for the training set $\{(\vec{x}_G, +)\} \cup \{(\vec{x}_{G'}, -) : G' \in \mathcal{G}'\}$
10:         compute a ranking $\pi_{G,i}$ of $\mathcal{C}$ from their "importance" in $F_i$
11:      compute an aggregated ranking $\pi_G$ of $\mathcal{C}$ from $\{\pi_{G,1}, \ldots, \pi_{G,t}\}$
12:      add $(G, \pi_G)$ to $\mathcal{D}$
13: $\Phi_{\text{FR}} \leftarrow$ learn a a feature ranking GNN model from $\mathcal{D}$
14: **return** $\Phi_{\text{FR}}$

---

**Global Graph Properties** These graph-level features provide high-level summaries of a graph's connectivity and topology. Examples of such global properties include the number of nodes and the average shortest path length.

Altogether, these three feature types yield 124 candidate features (see Table 1). Given the diversity of graph datasets, the importance of these features can vary significantly across graphs. The key challenge is to *automatically* identify a small subset for the task at hand that is most informative for the majority of training graphs.

## 3.2   Step 1: Training the FR-GNN Model

The first step of our method (see Algorithm 1 and the top part of Fig. 1) begins by selecting a random subset $\mathcal{D}_{\text{FR}} \subseteq \mathcal{D}_{\text{train}}$. In the experiments reported in Section 4, we consistently used 10% of the graphs in $\mathcal{D}_{\text{train}}$.

For each graph in $\mathcal{D}_{\text{FR}}$, the method computes all features from the pool $\mathcal{C}$ of candidate features. Node-level features are then aggregated into graph-level representations, as described in Section 3.1. In this way, each graph is mapped to a feature vector of dimension $|\mathcal{C}|$ (see the top-left matrix in Fig. 1 and lines 1–2 of Alg. 1).

To compute a feature ranking specific to each graph in $\mathcal{D}_{\text{FR}}$, we adopt a binary classification strategy based on random forests (see the top-right feature ranking matrix in Fig. 1 and lines 5–11 of Algorithm 1). More precisely, for all $G \in \mathcal{D}_{\text{FR}}$, we sample a set of graphs from $\mathcal{D}_{\text{FR}}$ that have a different target label than $G$. In our experiments, the number of these graphs randomly varies between 3 and 5. This sampling strategy ensures that the random forest learns to distinguish between graphs with different target labels, facilitating the identification of discriminative features. The training data for the random forest classifier consists of the graph-level feature vectors extracted from the sampled graphs, where the feature vector of $G$ is assigned the label '+', and those of the other graphs are assigned the label '−'. A random forest classifier with $T$ trees ($T = 128$ in the experiments) is then trained on this data, using $\sqrt{|\mathcal{C}|}$ candidate features at each split. Feature importance scores are computed using the Gini gain, which measures the total reduction in node impurity due to splits on that feature, normalized over all trees in the forest.

To ensure robustness, this procedure is repeated $t$ times ($t = 10$ in the experiments), each time with a newly sampled set of graphs from $\mathcal{D}_{\text{FR}}$. The final ranking $\pi_G$ for graph $G$ is obtained by aggregating the importance scores across the $t$ independent runs, resulting in a feature ranking for $G$ (see the top-right matrix in Fig. 1, which contains the individual rankings for each graph).

This process is repeated for all graphs in $\mathcal{D}_{\mathrm{FR}}$, producing a dataset, denoted $\mathcal{D}$ in the algorithm, that consists of training examples of the form $(G, \pi_G)$ for all $G \in \mathcal{D}_{\mathrm{FR}}$, where $\pi_G$ is the feature ranking computed for $G$ as described above. The algorithm then trains an FR-GNN on $\mathcal{D}$ and returns the resulting model $\Phi_{\mathrm{FR}}$ (see lines 13–14 of Algorithm 1 and the top-right box in Fig. 1).

*Remark* During preliminary experiments, we observed that the feature importance vectors produced by random forests often contain many values close to zero. These low scores can introduce unnecessary complexity into the $\Phi_{\mathrm{FR}}$ model. To address this, we introduce a dynamic thresholding strategy that adaptively identifies and removes low-importance features, retaining only those that make a clear contribution to the classification task. Specifically, for each graph, the feature importance values are first sorted in ascending order. We then compute the differences between consecutive values, defined as $\Delta_i = f_{i+1} - f_i$. If $\Delta_i$ exceeds the importance value $f_i$ itself, we set $f_i$ and all smaller values to zero. This adaptive mechanism effectively filters out features whose contribution is negligible, eliminating the need for a manually defined threshold. The remaining non-zero values correspond to features that exhibit a sufficiently sharp increase in importance relative to the preceding values, signaling their discriminative potential. These final rankings are then used as input to the $\Phi_{\mathrm{FR}}$ model.

### 3.3 Step 2: Training the GC-GNN Model

Given the feature learning model $\Phi_{\mathrm{FR}}$ obtained in the previous step, we apply it to each graph $G$ in the subset $\mathcal{D}_{\mathrm{GC}} = \mathcal{D}_{\mathrm{train}} \setminus \mathcal{D}_{\mathrm{FR}}$ to obtain a predicted feature ranking for $G$. These individual rankings are then aggregated to derive a global ranking of the candidate features. From this aggregated ranking, the top-$K$ features are selected and subsequently computed for the graphs in $\mathcal{D}_{\mathrm{GC}}$. We again stress that this is the step where the benefits of our method become apparent: it computes only the top-$K$ features from the pool $\mathcal{C}$ for each training graph in $\mathcal{D}_{\mathrm{GC}}$, rather than calculating all features in $\mathcal{C}$. Using the training set formed by the labeled graphs in $\mathcal{D}_{\mathrm{GC}}$, augmented with these top-$K$ features, the method trains a GC-GNN model to classify unseen graphs (see the bottom pipeline in Fig. 1). If the training graphs in $\mathcal{D}_{\mathrm{train}}$ are originally associated with node-level features, these features are used throughout the process and aggregated in the same way as the node-level features in $\mathcal{C}$ (see Section 3.1).

## 4 Experimental Evaluation

In this section, we experimentally evaluate the performance of the proposed method using real-world benchmark and synthetic datasets. The experiments are designed to address the following research questions:

**(Q1)** To what extent do features improve the predictive performance of GNNs for graph classification? If they do, how much improvement can be expected?

**(Q2)** How does the predictive performance of GC-GNN using the feature set selected by our method compare to using all features in the pool? How robust is this performance as the classification task becomes increasingly difficult?

**(Q3)** Are the $K$ features selected by our method genuinely relevant to the learning task, or does a randomly selected subset of $K$ features perform similarly?

### 4.1 Datasets

To address the above questions, we conducted experiments on a diverse collection of real-world and synthetic datasets. This section provides a detailed description of the datasets used.

*Real-World Datasets* To evaluate our method, we selected a diverse set of thirteen real-world graph classification benchmarks spanning the domains of molecular chemistry, bioinformatics, and computer vision. For molecular datasets, we considered MUTAG (Debnath et al., 1991; Kriege & Mutzel, 2012), BZR, DHFR, and COX2 (Sutherland et al., 2003), NCI1 and NCI-H23H (Wale et al., 2008; Kim et al., 2025; Shervashidze et al., 2011), PTCMM and PTCFR (Helma et al., 2001; Kriege & Mutzel, 2012). These datasets consist of graphs

where nodes represent atoms and edges denote chemical bonds, with target labels indicating biological activity. Specifically, MUTAG, BZR, DHFR, and COX2 focus on classifying molecules based on mutagenicity, activity against the benzodiazepine receptor, enzyme inhibition, and cyclooxygenase-2 activity, respectively. NCI1 targets activity against human non-small cell lung cancer, while PTCMM and PTCFR predict rodent carcinogenicity in male mice and female rats, respectively. The PROTEINS dataset (Borgwardt et al., 2005; Dobson & Doig, 2003) from bioinformatics consists of protein structures, where nodes represent secondary structure elements and edges denote spatial proximity between them. The classification task in PROTEINS is to distinguish enzymes from non-enzymes. Finally, the MSRC9 dataset (Neumann et al., 2016) from computer vision, derived from image segmentation, represents segmented images as graphs, with nodes and edges corresponding to superpixel relationships, and labels indicating object classes. We additionally evaluated our method on three social network datasets: Reddit-Binary, Reddit-Multi-5K, and Reddit-Multi-12K (Yanardag & Vishwanathan, 2015). In these datasets, nodes represent users, and an edge is created between two nodes if one responded to the other's comment. The task in Reddit-Binary is to classify each discussion graph as either a question/answer-based community or a discussion-based community. In Reddit-Multi-5K and Reddit-Multi-12K, the goal is to predict the subreddit category to which each discussion graph belongs. The three Reddit datasets, together with NCI-H23H, will also be used to investigate the scalability of our algorithm. Dataset statistics are summarized in Table 2 in Appendix B.

*Synthetic Datasets* To evaluate our method's ability to detect structural features defining target classes, we conducted experiments on synthetic graph datasets with known ground truth. For each learning problem defined below, we generated four datasets, each with 600 connected base graphs from one of four standard random graph models: Barabási–Albert (BA) (Barabási & Albert, 1999), Erdős–Rényi (ER) (Erdős & Rényi, 1959), Watts–Strogatz (WS) (Watts & Strogatz, 1998), and Power-Law (PL) cluster (Holme & Kim, 2002).

We defined the target classes using four distinct graphlet motifs: the 4-clique ($G_8$), the path $v_1 - v_2 - v_3 - v_4 - v_5$ on five vertices with chords $v_1 v_4$ and $v_2 v_5$ ($G_{20}$), the house graph ($G_{21}$), and the hexagon ($G_{30}$) (see Fig. 8 in Appendix A). They are pairwise topologically incomparable (i.e., none is subgraph isomorphic to any another), and the base graphs to which they are subsequently attached were generated to exclude all of them. Furthermore, the graphlets were attached in such a way that each graph satisfies exactly one class condition.

This design ensures that the presence of the graphlets acts as a discriminative signal rather than a background artifact. The four motifs are explicitly included in the candidate feature set, and the logical formulas defining each target class depend exclusively on the presence and interaction or absence of these motifs. This setup enables us to construct a hierarchy of graph classes through logical formulas of increasing structural complexity, all grounded in these four motifs. Importantly, it provides a controlled testbed for systematically evaluating the sensitivity and expressiveness of our feature selection mechanism. Specifically, if our method fails to distinguish classes that differ only in the combination of these known motifs, its limitations are revealed in a precise and interpretable manner.

For each base graph type, we considered five learning tasks, with target classes defined by the following logical formulas ($\overline{G}_i$ below denotes the absence of motif $G_i$):

$$
\begin{aligned}
\Phi_1 : &(\text{CLASS} = c_1 \iff G_8 \wedge \overline{G}_{20} \wedge \overline{G}_{21}) \wedge \\
&(\text{CLASS} = c_2 \iff \overline{G}_8 \wedge G_{20} \wedge \overline{G}_{21}) \wedge \\
&(\text{CLASS} = c_3 \iff \overline{G}_8 \wedge \overline{G}_{20} \wedge G_{21}) \wedge \\
&(\text{CLASS} = c_4 \iff \overline{G}_8 \wedge \overline{G}_{20} \wedge \overline{G}_{21})
\end{aligned}
\tag{1}
$$

$$
\begin{aligned}
\Phi_2 : &(\text{CLASS} = c_1 \iff G_8 \wedge G_{20} \wedge \overline{G}_{21}) \wedge \\
&(\text{CLASS} = c_2 \iff G_8 \wedge \overline{G}_{20} \wedge G_{21}) \wedge \\
&(\text{CLASS} = c_3 \iff \overline{G}_8 \wedge G_{20} \wedge G_{21})
\end{aligned}
\tag{2}
$$

$$
\begin{aligned}
\Phi_3 : &(\text{CLASS} = c \iff G_8 \wedge G_{20} \wedge G_{21}) \wedge \\
&(\text{CLASS} = \overline{c} \iff \overline{G}_8 \vee \overline{G}_{20} \vee \overline{G}_{21})
\end{aligned}
\tag{3}
$$

$$
\begin{aligned}
\Phi_4 : &(\text{CLASS} = c_0 \iff \overline{G}_8 \wedge \overline{G}_{20} \wedge \overline{G}_{21}) \wedge \\
&(\text{CLASS} = c_1 \iff (G_8 \wedge \overline{G}_{20} \wedge \overline{G}_{21}) \vee (\overline{G}_8 \wedge G_{20} \wedge \overline{G}_{21}) \vee (\overline{G}_8 \wedge \overline{G}_{20} \wedge G_{21})) \wedge \\
&(\text{CLASS} = c_2 \iff (G_8 \wedge G_{20} \wedge \overline{G}_{21}) \vee (G_8 \wedge \overline{G}_{20} \wedge G_{21}) \vee (\overline{G}_8 \wedge G_{20} \wedge G_{21})) \wedge \\
&(\text{CLASS} = c_3 \iff G_8 \wedge G_{20} \wedge G_{21})
\end{aligned}
\tag{4}
$$

$$
\begin{aligned}
\Phi_5 : &(\text{CLASS} = c_1 \iff (G_8 \vee G_{20}) \wedge G_{21} \wedge G_{30}) \wedge \\
&(\text{CLASS} = c_2 \iff (G_8 \vee G_{20} \vee G_{21}) \wedge \overline{G}_{30}) \wedge \\
&(\text{CLASS} = c_3 \iff \overline{G}_{21} \wedge G_{30}) \wedge \\
&(\text{CLASS} = c_4 \iff \overline{G}_8 \wedge \overline{G}_{20} \wedge ((G_{21} \wedge G_{30}) \vee (\overline{G}_{21} \wedge \overline{G}_{30})))
\end{aligned}
\tag{5}
$$

In $\Phi_1$, the four classes are pairwise disjoint and determined by the presence of exactly one motif or none, corresponding to a one-hot encoding over the three motifs. $\Phi_2$ defines three classes, each by the presence of exactly two motifs, making the problem more challenging due to feature overlaps. $\Phi_3$ is a concept learning task where the target class requires the presence of all three motifs, highlighting the difficulty for random forests with the Gini index, which evaluates features individually and is therefore less sensitive to feature interactions. In $\Phi_4$, four classes are defined by the presence of exactly $i$ motifs ($i = 0, 1, 2, 3$). $\Phi_5$ introduces a fourth motif and higher logical formula depth, testing robustness as class definitions become more complex.

## 4.2 Experimental Setting

For the experiments, we use three widely adopted GNN architectures: Graph Convolution Networks (GCN) (Kipf & Welling, 2016), Graph Attention Networks (GAT) (Velickovic et al., 2018), and Graph-SAGE (Hamilton et al., 2017). The dataset is split such that 10% of the graphs are used to train the FR-GNN, while the remaining 90% are further divided 70/30 for training and testing the GC-GNN. We report the average accuracy over 5 independent runs. For feature ranking, for each graph $G \in \mathcal{D}_{\text{FR}}$, $k \in \{3, 4, 5\}$ graphs with different labels are sampled. Feature importance is computed via a random forest with 128 trees and $\sqrt{|\mathcal{C}|}$ features per split.

All GNN models are optimized using Adam, with hyperparameters selected via grid search (learning rate of 0.005, dropout rate of 0.5). Experiments are run on an AMD Ryzen 9 5950X 16-core CPU @ 3.40GHz with 125 GB of memory. The PyTorch Geometric library (Fey & Lenssen, 2019) is used for data handling.

For all experiments, except those involving synthetic data with target classes defined by Eq. (5), we consistently used $K = 6$, i.e., selected the top six features. (We discuss the rationale for this choice in Sect. 4.3.1.) To address questions Q1–Q3, we compare the proposed method, denoted as T-6 (short for "top 6"), against three baselines: F-0: using no features (Q1), F-$\mathcal{C}$: using all features (Q2), and R-6: using six features selected

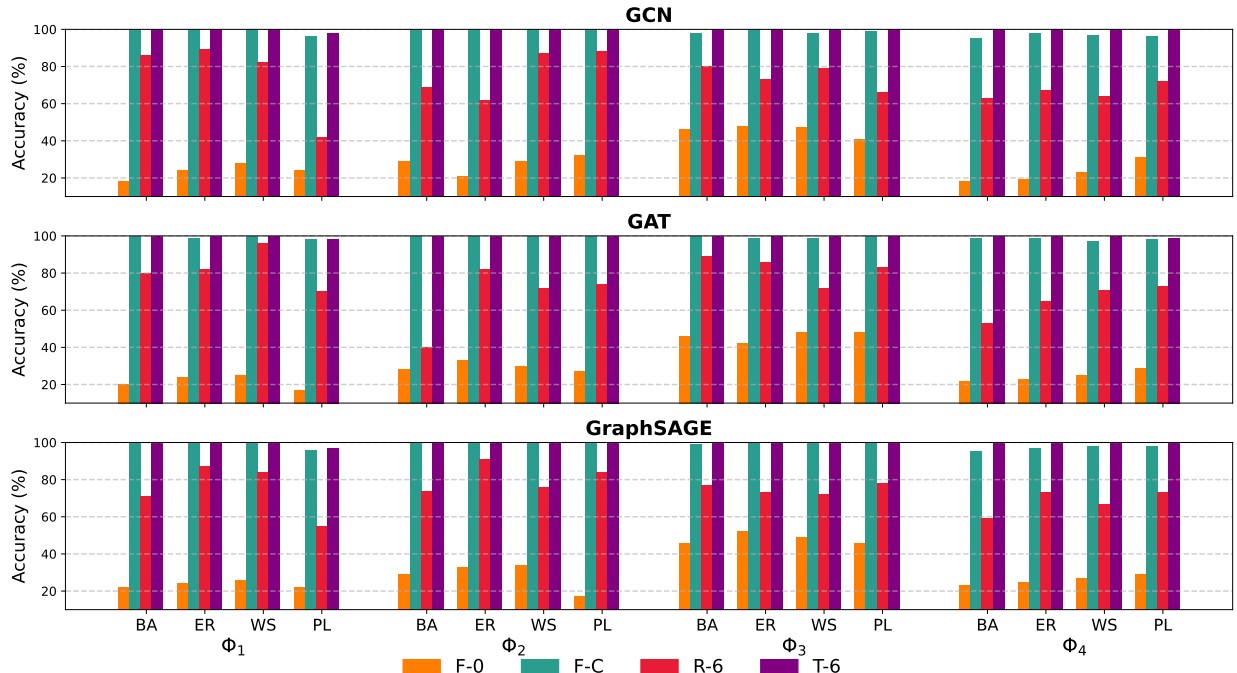

Figure 2: Grouped bar charts summarizing model performance on the synthetic datasets for the four classification tasks $\Phi_1$–$\Phi_4$. The top, middle, and bottom panels correspond to GCN, GAT, and GraphSAGE, respectively. Within each panel, datasets are grouped by task, and for each dataset, four color-coded bars represent the variants F-0, F-$\mathcal{C}$, R-6, and T-6.

at random from $\mathcal{C}$ (Q3). We refer to these as the "no-feature", "all-feature", and "random-feature" baselines, respectively.

### 4.3 Experimental Results

The results are presented as grouped bar charts in Figs. 2 and 3, and in Fig. 9 in Appendix A, for the synthetic and real-world datasets, across all three GNN architectures.

**Answer to Q1:** The results on the synthetic data provide a clear answer to this question (see Fig. 2 and Tables 4, 5, and 6 in Appendix B): baselines that utilize all features (F-$\mathcal{C}$), or even just six randomly selected features (R-6), significantly outperform the baseline that uses no features (F-0). Across the synthetic experiments, we observe improvements of up to 80% for F-$\mathcal{C}$ and up to 70% for R-6 relative to F-0.

The results for real-world datasets are more mixed (see Fig. 3 and Table 3 in Appendix B). For all three GNN architectures, F-0 is outperformed by both F-$\mathcal{C}$ and R-6 on some datasets (e.g., MUTAG), while on others (e.g., DHFR), it actually outperforms the other two baselines. These results suggest that using either too many features or a small number of randomly selected features can, in some cases, diminish predictive performance.

Furthermore, our method T-6 consistently outperforms F-0 across all real-world and synthetic datasets and all three GNN architectures, with average improvements of about at least 50% on the synthetic and about 14.2% for GCN, 15.8% for GAT, and 14.5% for GraphSAGE on the real-world datasets. On DHFR, where all three GNNs using no features perform better than when using either all or six random features, T-6 achieves a 6% improvement over F-0 with GCN. The improvement is even more dramatic on MUTAG, where T-6 outperforms F-0 by around 23% with GAT.

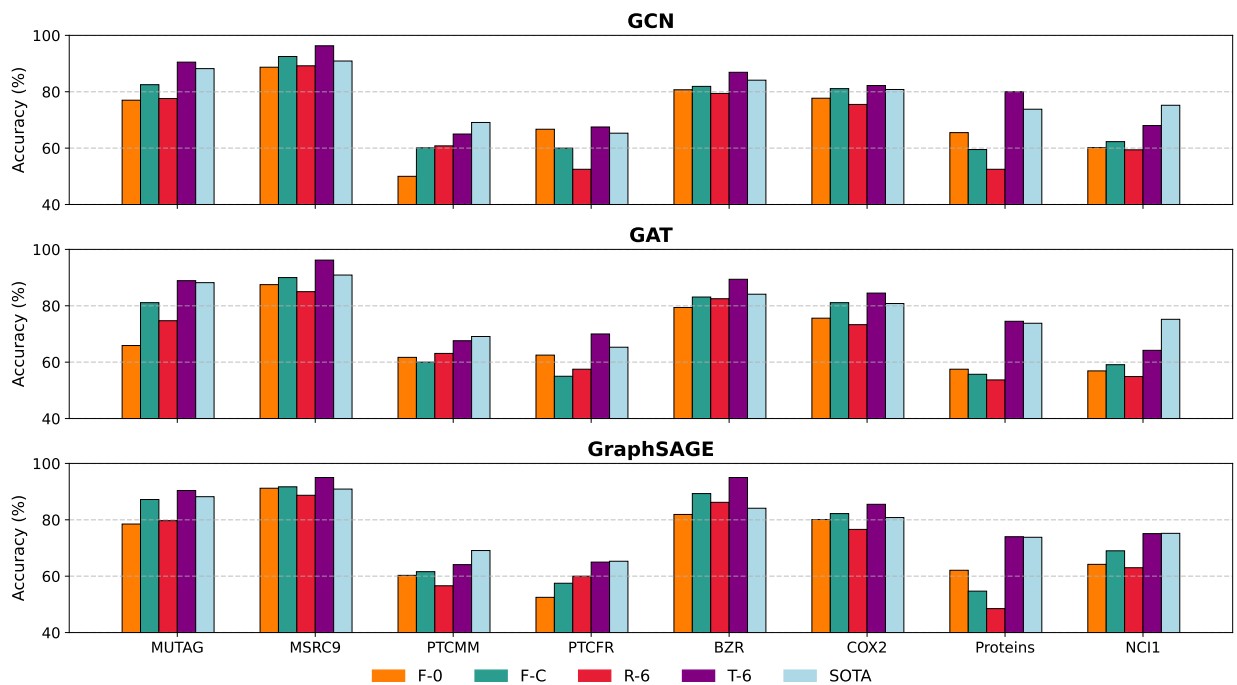

Figure 3: Grouped bar charts summarizing the results from Table 3 in Appendix B for the real-world datasets across three GNN architectures (top: GCN, middle: GAT, bottom: GraphSAGE). Each panel presents the accuracies across five variants — F-0, F-$\mathcal{C}$, R-6, T-6, and the reproduced SOTA results, shown as color-coded bars.

In summary, our answer to question Q1 is conditionally affirmative: features can substantially enhance predictive performance, as long as they are not overly numerous or randomly selected.

**Answer to Q2:** The accuracy results in Fig. 3 (see also Table 3 in Appendix B) clearly show that T-6 consistently outperforms F-$\mathcal{C}$ on real-world datasets across all three GNN architectures. Specifically, our method achieves average accuracies of 74.8% (GCN), 73.9% (GAT), and 75.1% (GraphSAGE) across 13 real-world benchmarks, compared to F-$\mathcal{C}$'s accuracies of 67.6% (GCN), 66.4% (GAT), and 69.0% (GraphSAGE). This corresponds to average improvements of 7.2%, 7.4%, and 6.2%, respectively. The most significant improvement, around 20% across all GNN architectures, was observed on the Proteins dataset. These results support our conjecture that too many features can lead to *overfitting*.

In contrast, on synthetic datasets (see Fig. 2 and Tables 4 and 5 in Appendix B), F-$\mathcal{C}$ achieves at least 90% accuracy on all five problems ($\Phi_1$–$\Phi_5$), across all four types of random graphs. Remarkably, for $\Phi_1$–$\Phi_4$, the six features selected by our method perform at least as well as F-$\mathcal{C}$, which attains nearly 96% accuracy, and sometimes even improve upon these results by up to 2%.

However, for $\Phi_5$ (see Fig. 9 and Table 5 in the Appendix), which involves four classes and more complex logical formulas, our method with six features achieves lower predictive performance than F-$\mathcal{C}$. Nevertheless, as the number of selected features increases, the performance difference becomes negligible (see the comparison between F-$\mathcal{C}$ and T-20).

In summary, GC-GNNs using the six features selected by our method consistently outperform those using all features on real-world datasets, demonstrating improved accuracy and reduced overfitting. On synthetic datasets, our method matches or exceeds the performance of using all features for simpler tasks and remains robust, with only a negligible performance gap as task complexity and the number of selected features increase.

**Answer to Q3:** Regarding the question of whether the six features selected by our method (T-6) are genuinely relevant to the learning task, we compared the accuracy results obtained by T-6 with those obtained using six features selected uniformly at random from the candidate pool. A closer look at Figs. 2 and 3 (see, also, Tables 3–5) reveals that T-6 consistently outperforms R-6 across all datasets and all three GNNs. In particular, on the real-world datasets, T-6 outperforms R-6 on average by 10.2% (GCN), 12.2% (GAT), and 12.3% (GraphSAGE). In some cases, the difference is dramatic (e.g., R-6 achieves 52.5% (GCN) on Proteins, while T-6 achieves 80%).

In summary, based on these results, we can provide a clear affirmative answer to Q3: The feature set returned by our method is genuinely relevant to the learning task and not selected by chance.

### 4.3.1  Further Results on the Real-World Data

We now present and discuss additional results obtained on the real-world benchmark datasets.

*Comparison to SOTA Results*  To contextualize the predictive performance of our approach, we compared it against existing SOTA results on the same real-world datasets. Specifically, we included only results that were published in peer-reviewed conference or journal papers, with publicly available code that could be executed on the corresponding datasets. Of the 13 benchmark datasets, eight met these criteria. For each of these eight datasets, we ran the respective algorithm and reported both the accuracy published in the literature and the accuracy achieved upon re-execution of the authors' code. In total, three recent algorithms (Akyol et al., 2021; Vincent-Cuaz et al., 2022; Wen et al., 2024) attained SOTA performance on at least one of these eight datasets.

On three of the eight datasets, our approach either outperforms or matches the accuracy *reported* in the literature within 2% (see the upper table in Table 3 in Appendix B). However, when comparing against results *reproduced* using the authors' publicly available code[2], our approach outperforms or closely matches the reproduced results (within 2%) on all eight datasets (including MUTAG, Proteins, NCI1 (Vincent-Cuaz et al., 2022), MSRC9 (Akyol et al., 2021), and PTCMM, PTCFR, BZR, and COX2 (Wen et al., 2024)).

*Statistical Significance*  We evaluated the statistical significance of the observed differences in predictive performance between T-6 and the three baselines (F-0, F-$\mathcal{C}$, and R-6) on the real-world datasets. For this purpose, we first applied the Kruskal-Wallis test. The null hypothesis (i.e., no performance differences) was rejected at $\alpha = 0.01$ (99% confidence level) with $p = 0.0016$, indicating strong evidence of performance variation across the methods. Given this result, we proceeded with Dunn's post-hoc test (Dunn, 1961) with Holm adjustment to identify pairwise differences. We found that T-6 significantly outperforms R-6 at the 99% confidence level, and achieves statistically significant improvements over both F-0 and F-$\mathcal{C}$ at the 95% confidence level. Thus, the improvements by T-6 are not only consistent but also statistically significant.

*Choice of $K$*  All experimental results presented in this paper, except those for the problems defined by $\Phi_5$, were obtained using $K = 6$. This value was selected by computing the average accuracy for all values of $K = 1, \ldots, 10$ and choosing the value (i.e., 6) that yielded the highest *average* performance. In Fig. 4 (left), we present not only the overall average accuracies but also the accuracy results for different $K$ values on the MUTAG and PTCMM datasets. While the optimal value of $K$ is 6 for MUTAG, it is 5 for PTCMM. We note that $K$ can alternatively be selected *automatically* for each dataset individually, by using a small validation subset of the training data and choosing the smallest value of $K$ that yields the highest predictive performance on this subset. Notice that, regardless of the technique used to select the optimal value of $K$ (e.g., via a validation set using Bayesian optimization), the following steps need to be performed only once: computing all features in $\mathcal{C}$ for the graphs in $\mathcal{D}_{\mathrm{FR}}$, training the random forest and the FR-GNN model, and predicting the feature ranking for the graphs in $\mathcal{D}_{\mathrm{GC}}$. Moreover, each feature considered for any value of $K$ must only be computed once for the graphs in $\mathcal{D}_{\mathrm{GC}}$. In contrast, the GC-GNN model must be retrained from scratch for every value of $K$ considered by the algorithm.

*Feature Analysis*  To further validate our feature selection approach, we conducted a qualitative analysis of several top-ranked features identified by our method for the real-world datasets. Specifically, we examined

---

[2]github.com/gamzeakyol/GNet for (Akyol et al., 2021), github.com/cedricvincentcuaz/TFGW for (Vincent-Cuaz et al., 2022), and github.com/TaoWen0309/TTG-NN for (Wen et al., 2024)

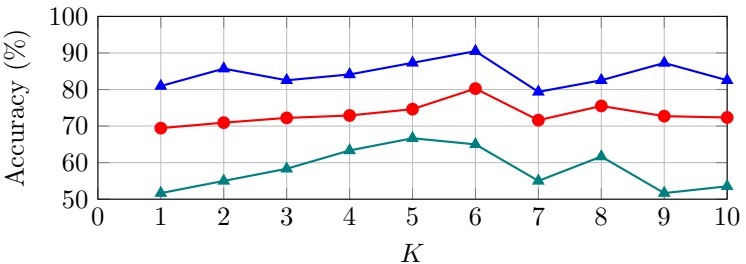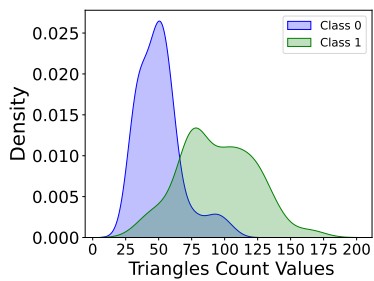

Figure 4: Left: Model accuracy of the top $K$ features selected by our method for different values of $K$ on two datasets: MUTAG (blue) and PTCMM (green), along with the average accuracy across all datasets using GCN (red). Right: Kernel density estimation (KDE) plots of the top-ranked feature for MUTAG (triangle count), showing the distribution of feature values across the two graph classes.

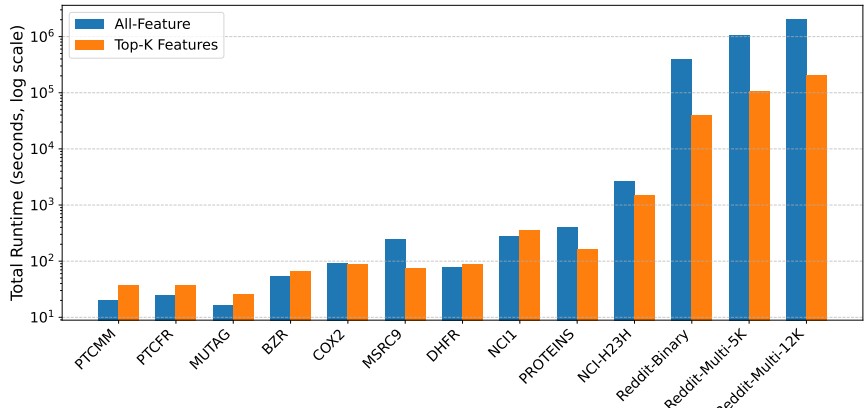

Figure 5: Total wall-clock runtime for each of the 13 benchmark datasets on both variants: the all-feature baseline and the proposed top-K method. Datasets are sorted in ascending size, defined as (number of graphs) × (average number of edges). The three large Reddit datasets, containing much larger graphs, are shown separately on the right and sorted by number of graphs to demonstrate scalability. Note the logarithmic scale on the $y$-axis.

the distributions of these features across graph classes using kernel density estimation (KDE), which allowed us to assess their discriminative power: The selected features exhibit meaningful and class-discriminative patterns. As an illustrative example, in the MUTAG dataset, the triangle count emerged as one of the top-ranked features (see Fig. 4, right). The KDE plot shows that class 1 graphs consistently exhibit higher triangle counts, suggesting that the triangle graphlet structure is a strong indicator of mutagenic activity and is effectively captured by our selection method.

*Empirical Runtime and Accuracy* Figure 5 presents the total runtime per dataset for both the all-feature baseline and our top-$K$ feature selection method (note the logarithmic scale on the $y$-axis). The datasets are sorted in ascending order according to the product of the number of graphs and the average number of edges.

While the runtime improvement is mixed across the first nine datasets, a clear advantage emerges for NCI-H23H (which contains many small graphs) and for the three Reddit datasets (which comprise increasing numbers of large graphs). For these four datasets, where the runtime of the all-feature baseline becomes particularly critical, our approach demonstrates a pronounced benefit. Specifically, it achieves up to a 90% reduction in total wall-clock time while simultaneously improving classification accuracy. The improvement is most pronounced for Reddit-Multi-12K, where the all-feature baseline required more than 557 hours,

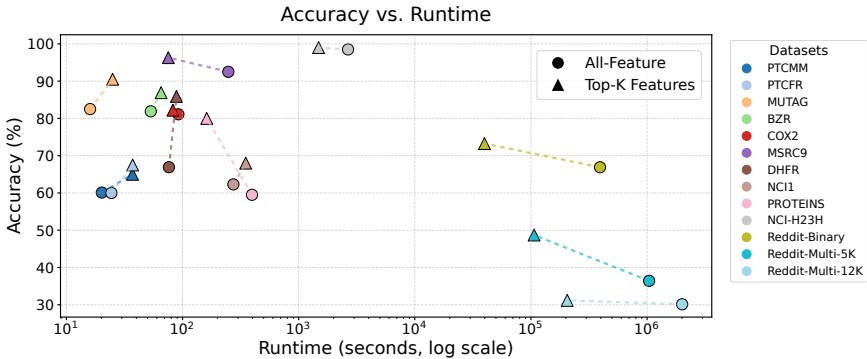

Figure 6: Accuracy–runtime comparison across 13 benchmark datasets. Each dataset is represented by a unique color, with circles denoting the all-feature baseline and triangles the proposed Top-K method. Dashed lines connect paired results for the same dataset. The x-axis shows total wall-clock runtime in seconds (log scale) and the y-axis shows classification accuracy.

whereas our approach completed the task in approximately 57 hours.

Importantly, this considerable reduction in computational time does not come at the expense of predictive performance; our method maintains, and in several cases improves, the classification accuracy compared to the all-feature baseline. The relationship between accuracy and total runtime is illustrated in Fig. 6. Our method achieves significantly lower runtimes on large-scale datasets while preserving or enhancing predictive performance. On smaller benchmarks, the feature ranking phase introduces a minor computational overhead; however, this cost is negligible compared to the consistent improvements in classification accuracy achieved by our method.

*Runtime Analysis* We now analyze the runtime required by our approach. Specifically, we compare the total time of the all-feature baseline ($\mathbb{T}_{\mathcal{C}}$) to that of our method ($\mathbb{T}_K$), and then proceed to compare the inference times of the two approaches.

Recall that the training data, $\mathcal{D}_{\text{train}}$, is partitioned into two subsets: $\mathcal{D}_{\text{FR}}$ and $\mathcal{D}_{\text{GC}}$, where $\mathcal{D}_{\text{FR}}$ (respectively, $\mathcal{D}_{\text{GC}}$) is used to train the FR-GNN (respectively, GC-GNN). Let $F_K$ denote the set of $K$ features selected by our method. The total training time for each approach can be expressed by

$$
\begin{aligned}
\mathbb{T}_{\mathcal{C}} &= \mathbb{T}_{\text{GC-GNN}}(\mathcal{D}_{\text{train}}, \mathcal{C}) \\
\mathbb{T}_K &= \mathbb{T}_{\text{FR-GNN}}(\mathcal{D}_{\text{FR}}, \mathcal{C}) + \mathbb{T}_{\text{GC-GNN}}(\mathcal{D}_{\text{GC}}, F_K) \ .
\end{aligned}
$$

Here, $\mathbb{T}_{\text{GC-GNN}}(\mathcal{D}_{\text{train}}, \mathcal{C})$ denotes the time required to compute all features in $\mathcal{C}$ for all graphs in $\mathcal{D}_{\text{train}}$ ($\mathbb{T}(\alpha_1)$), plus the time to train a GC-GNN using $\mathcal{D}_{\text{train}}$ with the resulting feature vectors ($\mathbb{T}(\alpha_2)$). The term $\mathbb{T}_{\text{FR-GNN}}(\mathcal{D}_{\text{FR}}, \mathcal{C})$ in $\mathbb{T}_K$ consists of the runtime of the following steps: Computing all features in $\mathcal{C}$ for all graphs in $\mathcal{D}_{\text{FR}}$ ($\mathbb{T}(\beta_1)$), training a random forest on the computed feature vectors and calculating a feature ranking for all graphs in $\mathcal{D}_{\text{FR}}$ ($\mathbb{T}(\beta_2)$), and training the FR-GNN on $\mathcal{D}_{\text{FR}}$ with the associated feature rankings ($\mathbb{T}(\beta_3)$). Finally, $\mathbb{T}_{\text{GC-GNN}}(\mathcal{D}_{\text{GC}}, F_K)$ in $\mathbb{T}_K$ includes the time to predict a feature ranking for each graph in $\mathcal{D}_{\text{GC}}$ using the FR-GNN model ($\mathbb{T}(\gamma_1)$), the time to determine the top $K$ features from these predicted rankings ($\mathbb{T}(\gamma_2)$), the time required to compute the $K$ features in $F_K$ for all graphs in $\mathcal{D}_{\text{GC}}$ ($\mathbb{T}(\gamma_3)$), and the time to train a GC-GNN using the resulting feature vectors ($\mathbb{T}(\gamma_4)$).

It follows that our approach achieves superior runtime performance compared to the all-feature baseline whenever $\mathbb{T}_{\mathcal{C}} > \mathbb{T}_K$, that is, if

$$
\mathbb{T}(\alpha_1) + \mathbb{T}(\alpha_2) > \mathbb{T}(\beta_1) + \mathbb{T}(\beta_2) + \mathbb{T}(\beta_3) + \mathbb{T}(\gamma_1) + \mathbb{T}(\gamma_2) + \mathbb{T}(\gamma_3) + \mathbb{T}(\gamma_4) \ . \tag{6}
$$

Using $\mathbb{T}(\alpha_2) \approx \mathbb{T}(\beta_3) + \mathbb{T}(\gamma_4)$ (follows from Blakely et al. (2021)) and the notation

$$
T = \mathbb{T}(\beta_2) + \mathbb{T}(\gamma_1) + \mathbb{T}(\gamma_2) \ ,
$$

(6) implies

$$\mathbb{T}(\alpha_1) > T + \mathbb{T}(\beta_1) + \mathbb{T}(\gamma_3) \ . \tag{7}$$

Consider the case that

$$T \leq \mathbb{T}(\beta_1) + \mathbb{T}(\gamma_3) \ . \tag{8}$$

Then (7) holds if

$$\mathbb{T}(\alpha_1) > 2\left(\mathbb{T}(\beta_1) + \mathbb{T}(\gamma_3)\right) \ . \tag{9}$$

Assuming that the *average* runtime to compute a feature from $\mathcal{C}$ for a graph in $\mathcal{D}_{\text{train}}$ equals that for features from $F_K$ in $\mathcal{D}_{\text{GC}}$ and for features from $\mathcal{C}$ in $\mathcal{D}_{\text{FR}}$, (9) can be expressed by

$$|\mathcal{C}| \cdot |\mathcal{D}_{\text{train}}| \cdot \bar{\mathbb{T}}_{\text{f}} > 2\left(|\mathcal{C}| \cdot |\mathcal{D}_{\text{FR}}| \cdot \bar{\mathbb{T}}_{\text{f}} + K \cdot |\mathcal{D}_{\text{GC}}| \cdot \bar{\mathbb{T}}_{\text{f}}\right) \ ,$$

which, in turn, holds whenever

$$\frac{|\mathcal{D}_{\text{GC}}|}{|\mathcal{D}_{\text{train}}|} > \frac{|\mathcal{C}|}{2(|\mathcal{C}| - K)} \tag{10}$$

because $\mathcal{D}_{\text{train}}$ is the disjoint union of $\mathcal{D}_{\text{FR}}$ and $\mathcal{D}_{\text{GC}}$.

In summary, under the above assumption, our approach achieves a better runtime than the all-feature baseline whenever both inequalities (8) and (10) hold. Specifically, (8) requires that the total time to (i) train the random forest, (ii) compute a feature ranking for each graph in $\mathcal{D}_{\text{FR}}$, (iii) predict feature rankings for all graphs in $\mathcal{D}_{\text{GC}}$ using the FR-GNN model, and (iv) select the top $K$ features from these predicted rankings is no greater than the time needed to compute all features for all graphs in $\mathcal{D}_{\text{FR}}$ plus the time to compute the $K$ features in $F_K$ for all graphs in $\mathcal{D}_{\text{GC}}$.

In Fig. 7(a), we present the three runtime terms from (8) for each real-world benchmark dataset analyzed in our experiments. Of the 13 datasets considered, (8) is satisfied by five: MSRC9, NCI-H23H, Reddit-Binary, Reddit-Multi-5K, and Reddit-Multi-12K. For these datasets, our approach is *faster* than the all-feature baseline (see Fig. 5), in accordance with (9), given that $\frac{|\mathcal{D}_{\text{GC}}|}{|\mathcal{D}_{\text{train}}|} \approx 0.86$ and $\frac{|\mathcal{C}|}{|\mathcal{C}| - K} \approx 1.05$ in all experiments. Notably, the four computationally expensive datasets (i.e., those requiring at least 1,000 seconds) are among this group, and the speed-up for the three Reddit datasets is at least one order of magnitude (note the log-scale on the $y$-axis).

We finish our runtime analysis by noting that our approach consistently offers an advantage in terms of *inference time*. Specifically, for each graph to be classified, it requires computing only $K$ features from the full candidate set $\mathcal{C}$. This is particularly important for applications in which the GC-GNN model must be applied to a (potentially very) large number of graphs.

*Scalability* To assess scalability, we extended our experimental evaluation to include four larger benchmarks: Reddit-Binary, Reddit-Multi-5K, Reddit-Multi-12K, and NCI-H23H, with dataset statistics summarized in Table 2 in Appendix B. The three Reddit datasets each comprise graphs of similar size, whereas the average graph size in NCI-H23H is approximately one order of magnitude smaller.

We first investigate how our approach scales with increasing average graph size. For the Reddit datasets, which all contain large graphs, the per-graph runtime of the all-feature baseline ranges from 168 to 208 seconds, while our method achieves 17 to 21 seconds per graph—an order-of-magnitude reduction. On the large NCI-H23H dataset, containing many small graphs, both methods yield very low per-graph runtimes (0.07 seconds for the baseline vs. 0.04 seconds for our method), but our method remains more efficient.

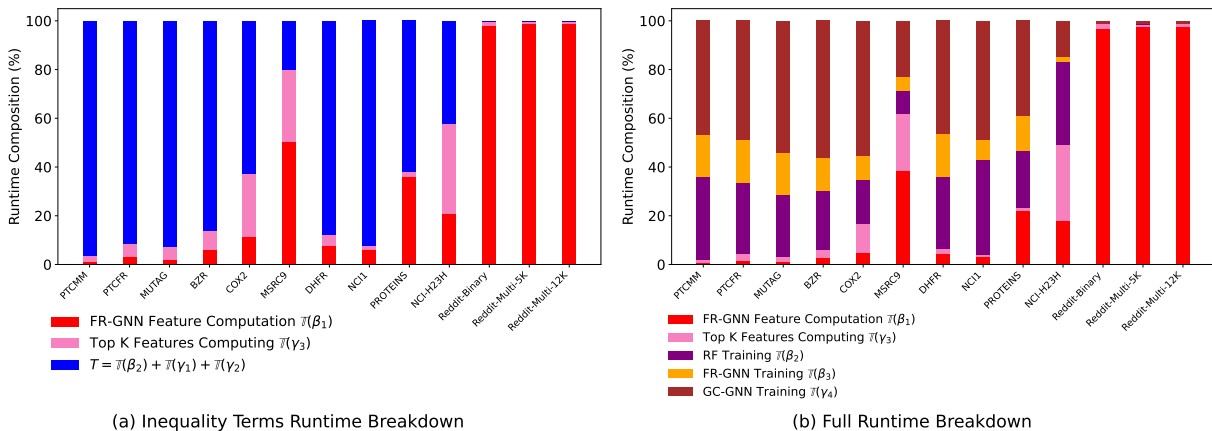

Figure 7: Normalized percentage breakdown of running time per dataset. The stacked bars show the relative contribution of different computational components. (a) Breakdown corresponding to the terms defined in inequality (8). (b) Full decomposition of the end-to-end pipeline, including all components.

Thus, for our method, increasing the average graph size by one order of magnitude results in a roughly two-order-of-magnitude increase in per-graph runtime.

This behavior is expected, since the candidate feature set includes computationally expensive operations (e.g., graphlet counts). A closer examination of Fig. 7 confirms that, for the three Reddit datasets, the time required to compute the full feature set for the graphs in $\mathcal{D}_{\mathrm{FR}}$ dominates the total runtime.

When considering only the three Reddit datasets, which contain graphs of similar size, the per-dataset computational time scales linearly with the number of graphs. Indeed, the per-graph computational time is nearly equal for these datasets (Reddit-Binary: 19.40 sec, Reddit-Multi-5K: 21.31 sec, Reddit-Multi-12K: 17.22 sec). This indicates that increasing database cardinality alone does not significantly affect the per-graph computational effort.

In summary, the primary determinant of per-graph runtime is the average graph size. This implies that *for (very) large graphs, the time complexity of computing a candidate feature should be quasilinear in the number of vertices.*

### 4.3.2 Further Results on the Synthetic Data

*Other Feature Ranking Approaches* We evaluated the performance of three methods: random forests (RF), support vector machines (SVM) (Cortes & Vapnik, 1995), and neural networks coupled with the LIME local explainer (Ribeiro et al., 2016) (NN+LIME), in identifying key graphlet features across the five learning problems $\Phi_1$–$\Phi_5$ defined in Section 4.1. RF consistently identified the three key graphlets ($G_8$, $G_{20}$, and $G_{21}$) among the top-ranked features for the problems defined by $\Phi_1$–$\Phi_4$. Notably, other features, such as the total number of cliques and the node clique number, were also frequently selected as important. Upon investigation, we found these features to be semantically aligned with the identified graphlets, particularly since $G_8$ corresponds to a 4-clique. SVM exhibited similar, though slightly less consistent, behavior, with the key graphlets appearing lower in the ranking compared to RF. In contrast, NN+LIME could not consistently identify the relevant graphlets. For problem $\Phi_5$, all three methods struggled to consistently rank the four relevant graphlets among the top features. However, RF demonstrated comparatively stronger performance, as the four features associated with $\Phi_5$ were ranked higher than by SVM and NN+LIME.

We also compared the runtime of these three algorithms. NN+LIME required the longest training time, followed by RF, while SVM was the fastest but less accurate in identifying the most informative features. Since accuracy was our primary concern, we chose RF for all subsequent experiments.

*The Count-Based Problem* We also explored a learning problem (denoted $\Phi_6$) where the class label was determined by the *number* of graphlets from $G_8$, $G_{20}$, and $G_{21}$ present in the graph. Specifically, graphs

containing at most one motif from this set were assigned to class 0, those containing two to four motifs to class 1, and graphs with five or more motifs to class 2. The results exhibited behavior similar to that observed in learning problems $\Phi_1$–$\Phi_4$ (see Fig. 9 and Table 6 in the Appendix).

## 5 Concluding Remarks

We have proposed a method that automatically selects a small subset of relevant features from a large pool of candidate features to improve the predictive performance of GNNs for graph classification. Our experimental results on thirteen nine benchmark datasets and three GNN architectures clearly demonstrate that: (1) incorporating node features can significantly enhance predictive performance compared to vanilla GNNs, (2) to prevent overfitting, it is crucial to select a small, informative subset of features from a large candidate pool rather than using all available features, and (3) the features selected by our method are genuinely relevant to the learning task, rather than being chosen by chance.

We also evaluated the relevance of the selected features in a more controlled and precise setting. Specifically, we generated synthetic base graphs and defined target classes using logical formulas of increasing complexity, based on the presence or absence of a small set of motifs. Our results show that the approach can accurately identify relevant features and successfully learn these classification tasks. However, as the complexity of the formulas increases, more features must be selected to maintain high accuracy.

While our method demonstrates effective feature selection for improving GNN-based graph classification, several promising avenues remain for future exploration. First, scaling the approach to handle very large graphs is a natural next step. In our experiments, the FR-GNN was trained separately for each dataset. We also explored training a single, global FR-GNN across all datasets. Although this global model led to GC-GNNs with higher average accuracy than the three baselines, its performance gains were less pronounced than those achieved by individually trained FR-GNNs (see Table 7 in the Appendix). Thus, another question is how to adapt and extend our approach to train a global FR-GNN that can produce task-specific GC-GNNs with predictive performance at least comparable to those reported here. Additionally, investigating the robustness of our method to noisy or adversarial features, developing theoretical guarantees for feature subset optimality, and establishing bounds on generalization error or subset optimality would further deepen the understanding and reliability of our approach. Finally, it would be useful to modify the method to account for feature computation time, selecting the most informative features with the smallest cumulative runtime.

### Broader Impact Statement

Our framework automates feature subset selection for graph classification by choosing a small, task-specific subset from a user-defined universe of candidate features. This improves predictive performance and reduces feature computation time. However, the method does not scale well to very large graphs, limiting applicability in such settings. The key ethical consideration is in defining the universal candidate feature set, which requires careful design by domain experts to avoid bias and ensure relevance. Transparent reporting and validation are necessary, especially for sensitive or high-stakes applications. By balancing expert knowledge, automation, and efficiency, our approach supports responsible and interpretable AI for graph-structured data.

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

# A Appendix: Figures

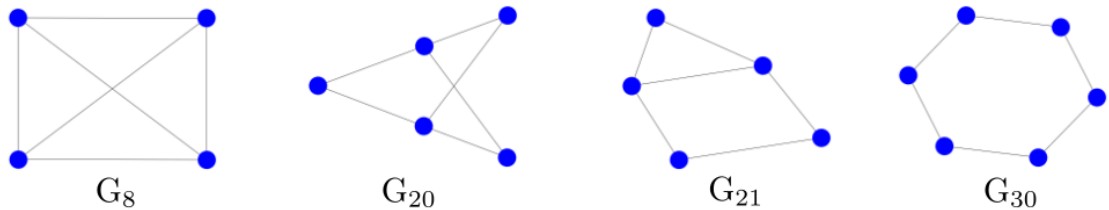

Figure 8: The four graphlets used in our experiments. They are pairwise incomparable (i.e., none is subgraph isomorphic to another).

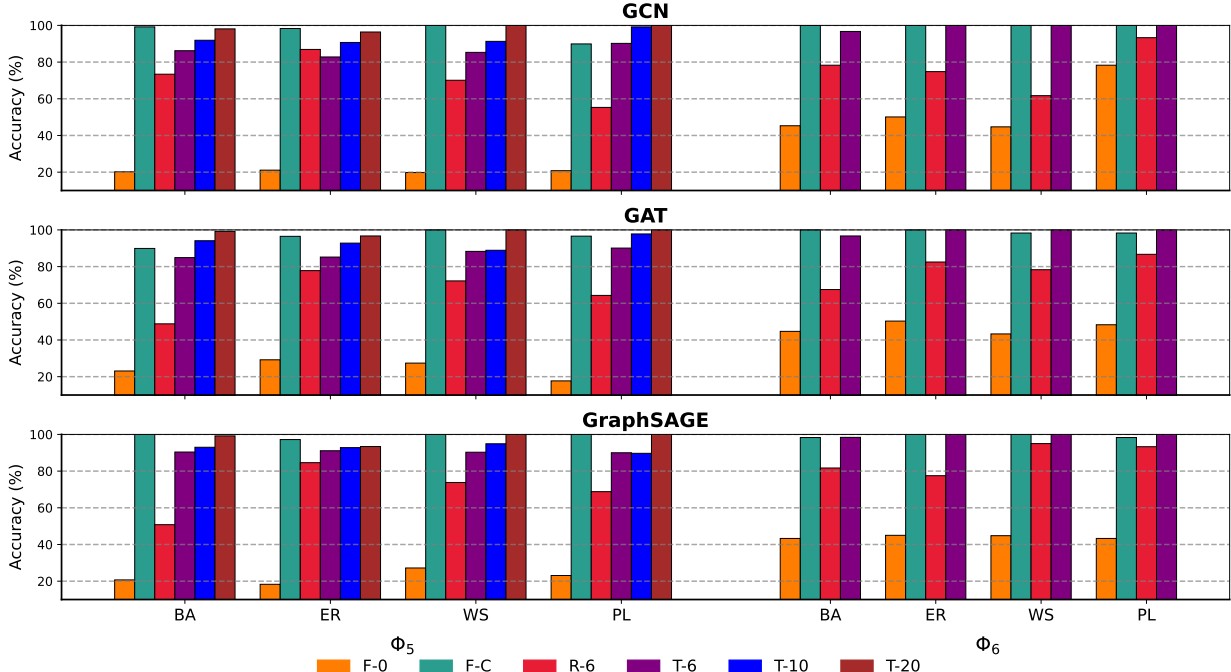

Figure 9: Grouped bar charts summarizing the results from Table 5 and Table 6 in Appendix B for the classification tasks $\Phi_5$ (left) and the count-based learning problem $\Phi_6$ across three GNN architectures (top: GCN, middle: GAT, bottom: GraphSAGE).

# B    Appendix: Tables

Table 2: Summary of datasets used in our experiments. For each dataset, we provide the number of graphs, average nodes and edges per graph, number of node attributes, and number of graph classes.

| dataset | #graphs | avg. #nodes | avg. #edges | #attributes | #classes |
|---|---|---|---|---|---|
| MUTAG | 188 | 17.90 | 39.60 | 7 | 2 |
| BZR | 405 | 35.75 | 38.36 | 53 | 2 |
| DHFR | 756 | 42.43 | 44.54 | 53 | 2 |
| COX2 | 467 | 41.22 | 43.45 | 35 | 2 |
| NCI1 | 4110 | 29.87 | 32.30 | 37 | 2 |
| PTCMM | 336 | 13.97 | 14.32 | 20 | 2 |
| PTCFR | 351 | 14.56 | 15.00 | 19 | 2 |
| Proteins | 1113 | 39.10 | 145.60 | 3 | 2 |
| MSRC9 | 221 | 40.58 | 97.94 | 10 | 8 |
| NCI-H23H | 40 353 | 46.67 | 48.69 | 0 | 2 |
| Reddit-Binary | 2000 | 429.63 | 497.75 | 0 | 2 |
| Reddit-Multi-5K | 4999 | 508.52 | 594.87 | 0 | 5 |
| Reddit-Multi-12K | 11 929 | 391.41 | 456.89 | 0 | 11 |

Table 3: Classification accuracy (%) on real-world datasets using GCN, GAT, and GraphSAGE architectures. Results are shown for our method (T-6) and the baselines (F-0, F-$\mathcal{C}$, and R-6). The best result per dataset within each GNN architecture is indicated by ✓ (GCN), † (GAT), and ‡ (GraphSAGE); the overall best result per dataset across all three architectures is highlighted by a box. SOTA results from the literature are provided for each dataset (second-to-last row of the upper table). Specifically, MUTAG, Proteins and NCI1 results from (Vincent-Cuaz et al., 2022), MSRC9 from (Akyol et al., 2021), PTCMM, PTCFR, BZR, and COX2 from (Wen et al., 2024). Only peer-reviewed conference and journal publications have been considered. Where possible, we reproduced results using the authors' code (last row of the upper table). The bottom table reports results for datasets where reproducing the SOTA results was not possible.

| | MUTAG | MSRC9 | PTCMM | PTCFR | BZR | COX2 | Proteins | NCI1 |
|---|---|---|---|---|---|---|---|---|
| | | | | GCN | | | | |
| F-0 | 77.0±1.1 | 88.7±5.3 | 50.0±4.7 | 66.7±3.7 | 80.7±1.1 | 77.7±2.2 | 65.5±4.9 | 60.2±0.6 |
| F-$\mathcal{C}$ | 82.5±2.2 | 92.5±3.5 | 60.1±4.7 | 60.0±3.8 | 81.9±0.9 | 81.1±1.5 | 59.5±9.3 | 62.3±1.1 |
| R-6 | 77.6±1.3 | 89.2±4.6 | 60.8±5.9 | 52.5±3.2 | 79.4±0.9 | 75.5±1.4 | 52.5±8.8 | 59.4±0.3 |
| T-6 | 90.5±1.1✓ | 96.3±1.8✓ | 65.0±2.4✓ | 67.5±2.6✓ | 86.9±0.9✓ | 82.2±3.2✓ | 80.0±5.7✓ | 68.0±1.1✓ |
| | | | | GAT | | | | |
| F-0 | 65.9±5.6 | 87.5±3.5 | 61.7±2.3 | 62.5±2.4 | 79.4±2.7 | 75.6±1.8 | 57.5±4.9 | 56.9±1.9 |
| F-$\mathcal{C}$ | 81.1±3.6 | 90.0±7.1 | 60.0±9.4 | 55.0±7.6 | 83.1±4.4 | 81.1±3.3 | 55.7±9.9 | 59.1±1.7 |
| R-6 | 74.7±1.3 | 85.0±0.2 | 63.1±0.3 | 57.5±3.3 | 82.5±2.4 | 73.3±2.5 | 53.7±9.8 | 54.9±5.8 |
| T-6 | 88.9±1.8† | 96.2±5.3† | 67.6±3.5† | 70.0±3.2† | 89.4±0.9† | 84.5±1.2† | 74.5±2.1† | 64.2±3.4† |
| | | | | GraphSAGE | | | | |
| F-0 | 78.5±3.3 | 91.2±1.7 | 60.3±5.1 | 52.5±4.5 | 81.9±6.2 | 80.1±4.4 | 62.1±0.1 | 64.2±1.1 |
| F-$\mathcal{C}$ | 87.2±0.7 | 91.7±1.0 | 61.6±2.3 | 57.5±1.9 | 89.3±4.4 | 82.2±2.2 | 54.7±9.3 | 69.0±0.4 |
| R-6 | 79.7±1.6 | 88.7±5.3 | 56.6±0.8 | 60.0±0.9 | 86.2±1.7 | 76.6±1.4 | 48.5±6.3 | 63.0±0.7 |
| T-6 | 90.4±1.4‡ | 95.0±1.1‡ | 64.1±1.2‡ | 65.0±1.4‡ | 95.0±1.2‡ | 85.5±2.4‡ | 74.0±2.8‡ | 75.1±0.5‡ |
| | | | | SOTA Results | | | | |
| reported (paper) results | 96.4±3.3 | 95.4 | 74.1±4.6 | 73.2±3.9 | 87.4±2.6 | 86.7±3.4 | 82.9±2.7 | 88.1±2.5 |
| reproduced (our run) results | 88.2±1.5 | 90.9±3.4 | 69.1±2.1 | 65.3±2.0 | 84.1±1.7 | 80.8±1.3 | 73.8±4.7 | 75.2±0.8 |

| | DHFR | NCI-H23H | Reddit-Binary | Reddit-Multi-5K | Reddit-Multi-12K |
|---|---|---|---|---|---|
| | | | GCN | | |
| F-0 | 79.5±0.3 | 50.9 ± 1.3 | 49.8 ± 2.4 | 19.1 ± 1.4 | 21.8 ± 2.1 |
| F-$\mathcal{C}$ | 66.9±2.6 | 98.5 ± 0.8 | 66.9 ± 1.9 | 36.4 ± 2.5 | 30.1 ± 1.4 |
| R-6 | 76.9±0.9 | 91.8 ± 2.6 | 61.7 ± 4.5 | 40.8 ± 3.7 | 22.0 ± 3.3 |
| T-6 | 85.9±0.5✓ | 99.2 ± 0.7✓ | 73.3 ± 2.0✓ | 48.7 ± 2.1✓ | 31.2 ± 2.2✓ |
| | | | GAT | | |
| F-0 | 70.9±9.1 | 51.1 ± 2.1 | 46.2 ± 1.4 | 19.3 ± 1.1 | 20.5 ± 1.8 |
| F-$\mathcal{C}$ | 61.6±7.5 | 98.6 ± 1.0 | 72.5 ± 2.4 | 37.4 ± 2.6 | 28.4 ± 1.6 |
| R-6 | 60.3±4.0 | 68.2 ± 4.6 | 67.7 ± 3.7 | 37.2 ± 3.0 | 23.5 ± 2.7 |
| T-6 | 75.3±1.3† | 99.1 ± 0.9† | 73.3 ± 2.8 † | 46.9 ± 2.1† | 30.5 ± 1.4† |
| | | | GraphSAGE | | |
| F-0 | 77.1±5.0 | 50.8 ± 1.6 | 49.5 ± 2.9 | 19.7 ± 1.5 | 20.7 ± 1.9 |
| F-$\mathcal{C}$ | 66.2±5.3 | 98.6 ± 1.2 | 73.2 ± 2.0 | 35.8 ± 2.7 | 29.5 ± 2.0 |
| R-6 | 74.3±0.8 | 64.3 ± 3.8 | 66.4 ± 3.5 | 29.2 ± 4.5 | 23.4 ± 2.4 |
| T-6 | 79.6±6.6‡ | 99.1 ± 0.8‡ | 80.5 ± 2.7‡ | 42.6 ± 1.6‡ | 31.0 ± 1.5‡ |

Table 4: Classification accuracy (%) of the four classification tasks defined by Eq. (1)–(4) on synthetic datasets generated using four graph models: Barabási–Albert (BA), Erdős–Rényi (ER), Watts–Strogatz (WS), and the Power-Law (PL) Cluster. Bold indicates the best result per graph model within each GNN architecture, while underlining highlights the overall best performance for each task.

| | GCN | | | | GAT | | | | GraphSAGE | | | |
|---|---|---|---|---|---|---|---|---|---|---|---|---|
| | F-0 | F-$\mathcal{C}$ | R-6 | T-6 | F-0 | F-$\mathcal{C}$ | R-6 | T-6 | F-0 | F-$\mathcal{C}$ | R-6 | T-6 |
| $\Phi_1$ defined by Eq. (1) | | | | | | | | | | | | |
| BA | 18.2±7.1 | **100**±0.0 | 86.1±1.2 | **100**±0.0 | 19.9±6.4 | **100**±0.0 | 80.3±1.1 | **100**±0.0 | 22.4±5.3 | **100**±0.0 | 70.6±2.1 | **100**±0.0 |
| ER | 24.3±2.4 | **100**±0.0 | 88.5±2.8 | **100**±0.0 | 24.3±3.2 | 98.9±0.3 | 82.1±2.5 | **100**±0.0 | 24.1±1.6 | **100**±0.0 | 87.0±2.5 | **100**±0.0 |
| WS | 27.6±7.8 | **100**±0.0 | 82.2±6.4 | **100**±0.0 | 25.4±6.9 | **100**±0.0 | 96.2±1.1 | **100**±0.0 | 26.3±4.5 | **100**±0.0 | 84.1±4.1 | **100**±0.0 |
| PL | 24.3±3.5 | 96.2±0.8 | 42.1±9.9 | **98.3**±1.2 | 17.3±2.3 | **98.2**±1.0 | 69.8±5.6 | 98.1±0.3 | 21.8±3.2 | 95.9±0.9 | 55.3±3.2 | **97.3**±1.1 |
| $\Phi_2$ defined by Eq. (2) | | | | | | | | | | | | |
| BA | 29.1±2.1 | **100**±0.0 | 69.0±7.7 | **100**±0.0 | 27.8±3.6 | **100**±0.0 | 40.1±4.8 | **100**±0.0 | 29.1±2.4 | **100**±0.0 | 73.6±3.3 | **100**±0.0 |
| ER | 20.8±11.3 | **100**±0.0 | 61.8±3.6 | **100**±0.0 | 33.3±7.4 | **100**±0.0 | 82.3±2.7 | **100**±0.0 | 32.8±5.6 | **100**±0.0 | 90.5±3.9 | **100**±0.0 |
| WS | 28.9±4.2 | **100**±0.0 | 87.4±4.8 | **100**±0.0 | 29.7±2.1 | **100**±0.0 | 72.2±2.3 | **100**±0.0 | 34.2±3.9 | **100**±0.0 | 76.3±3.2 | **100**±0.0 |
| PL | 31.7±1.7 | **100**±0.0 | 88.1±1.3 | **100**±0.0 | 27.3±1.8 | **100**±0.0 | 73.6±1.3 | **100**±0.0 | 17.3±2.1 | **100**±0.0 | 84.2±1.7 | **100**±0.0 |
| $\Phi_3$ defined by Eq. (3) | | | | | | | | | | | | |
| BA | 46.2±2.1 | 98.2±0.6 | 80.1±5.7 | **100**±0.0 | 45.9±1.8 | **100**±0.0 | 89.4±4.5 | **100**±0.0 | 46.3±2.7 | 99.0±0.1 | 77.0±2.5 | **100**±0.0 |
| ER | 48.3±3.5 | **100**±0.0 | 73±0.7 | **100**±0.0 | 42.2±3.2 | 99.1±0.2 | 86.1±3.6 | **100**±0.0 | 51.5±2.4 | **100**±0.0 | 72.8±2.9 | **100**±0.0 |
| WS | 46.7±1.7 | 98.3±0.7 | 79.1±2.1 | **100**±0.0 | 48.2±2.2 | 99.0±0.2 | 71.9±3.2 | **100**±0.0 | 48.6±3.9 | **100**±0.0 | 71.9±3.0 | **100**±0.0 |
| PL | 40.9±1.9 | 99.1±0.2 | 66.2±1.6 | **100**±0.0 | 47.8±3.7 | **100**±0.0 | 82.8±2.8 | **100**±0.0 | 46.4±1.6 | **100**±0.0 | 77.8±2.4 | **100**±0.0 |
| $\Phi_4$ defined by Eq. (4) | | | | | | | | | | | | |
| BA | 17.8±1.4 | 94.8±1.1 | 62.8±3.6 | **100**±0.0 | 22.0±1.3 | 97.4±0.2 | 52.7±2.9 | **100**±0.0 | 23.3±3.1 | 95.0±0.8 | 58.7±6.3 | **100**±0.0 |
| ER | 18.8±8.4 | 97.7±0.4 | 66.6±3.5 | **100**±0.0 | 22.8±1.5 | 98.0±0.1 | 65.0±3.1 | **100**±0.0 | 25.2±2.0 | 97.1±1.1 | 72.9±4.1 | **100**±0.0 |
| WS | 22.9±5.3 | 97.1±0.6 | 63.9±2.7 | **100**±0.0 | 24.9±2.1 | 96.6±0.3 | 71.3±2.6 | **100**±0.0 | 26.8±2.1 | 97.2±0.5 | 67.3±2.9 | **100**±0.0 |
| PL | 30.9±2.1 | 96.2±1.3 | 72.1±1.9 | **100**±0.0 | 29.4±2.3 | 98.2±0.2 | 73.1±2.1 | **99.1**±0.2 | 29.1±3.1 | 98.2±0.2 | 73.2±2.3 | **100**±0.0 |

Table 5: Classification accuracy (%) of the classification task defined by Eq. (5) on synthetic datasets generated using four graph models: Barabási–Albert (BA), Erdős–Rényi (ER), Watts–Strogatz (WS), and Power-Law (PL) Cluster. Bold indicates the best result per graph model within each GNN architecture, while underlining highlights the overall best performance.

| | $\Phi_5$ defined by Eq. (5) | | | |
| | BA | ER | WS | PL |
|---|---|---|---|---|
| | GCN | | | |
| F-0 | 20.2±3.4 | 21.1±4.1 | 19.8±2.7 | 20.8±2.3 |
| F-$\mathcal{C}$ | **99.2**±0.1 | **98.3**±0.2 | **100**±0.0 | 89.9±0.2 |
| R-6 | 73.4±4.7 | 86.9±2.9 | 70.1±3.9 | 55.3±2.4 |
| T-6 | 86.2±2.3 | 82.8±3.4 | 85.3±2.1 | 90.2±1.1 |
| T-10 | 91.9±1.1 | 90.7±1.3 | 91.3±1.4 | 99.2±0.3 |
| T-20 | 98.1±0.2 | 96.4±0.7 | **100**±0.0 | **100** ±0.0 |
| | GAT | | | |
| F-0 | 23.1±3.2 | 29.2±3.1 | 27.4±1.6 | 17.7±1.9 |
| F-$\mathcal{C}$ | 89.9±0.4 | 96.5±0.2 | **100**±0.0 | 96.6±0.4 |
| R-6 | 48.8±3.1 | 77.8±1.4 | 72.2±1.6 | 64.3±2.7 |
| T-6 | 84.9±2.6 | 85.2±1.8 | 88.3±2.0 | 90.1±1.7 |
| T-10 | 94.1±1.5 | 92.8±1.3 | 88.9±1.6 | 97.8±0.3 |
| T-20 | **99.2**±0.2 | **96.7**±0.2 | **100**±0.0 | **100**±0.0 |
| | GraphSAGE | | | |
| F-0 | 20.7±2.0 | 18.3±2.6 | 27.2±1.4 | 23.1±1.5 |
| F-$\mathcal{C}$ | **100**±0.0 | **97.2**±0.6 | **100**±0.0 | **100**±0.0 |
| R-6 | 50.8±2.6 | 84.6±2.9 | 73.8±3.1 | 68.8±2.0 |
| T-6 | 90.4±2.5 | 91.1±3.1 | 90.3±2.8 | 90.0±1.1 |
| T-10 | 93.0±1.2 | 92.8±2.7 | 94.9±0.7 | 89.7±0.2 |
| T-20 | 99.2±0.2 | 93.4±1.4 | **100**±0.0 | **100**±0.0 |

Table 6: Classification accuracy (%) of the count-based learning problem on synthetic datasets generated using four graph models: Barabási–Albert (BA), Erdős–Rényi (ER), Watts–Strogatz (WS), and Power-Law (PL) Cluster. Bold indicates the best result per graph model within each GNN architecture, while underlining highlights the overall best performance.

| | $\Phi_6$ (the count-based problem) | | | |
| | BA | ER | WS | PL |
|---|---|---|---|---|
| | GCN | | | |
| F-0 | 45.3±1.7 | 50.1±3.3 | 44.7±1.9 | 78.3±3.5 |
| F-$\mathcal{C}$ | **100**±0.0 | **100**±0.0 | **100**±0.0 | **100**±0.0 |
| R-6 | 78.3±2.9 | 74.8±1.8 | 61.7±2.0 | 93.3±2.1 |
| T-6 | 96.7±1.1 | **100**±0.0 | **100**±0.0 | **100**±0.0 |
| | GAT | | | |
| F-0 | 44.7±2.6 | 50.3±1.4 | 43.3±1.9 | 48.3±1.5 |
| F-$\mathcal{C}$ | **100**±0.0 | **100**±0.0 | 98.3±0.5 | 98.3±0.4 |
| R-6 | 67.5±4.4 | 82.5±2.1 | 78.3±0.9 | 86.7±1.5 |
| T-6 | 96.7±1.2 | **100**±0.0 | **100**±0.0 | **100**±0.0 |
| | GraphSAGE | | | |
| F-0 | 43.3±1.7 | 45.0±1.3 | 44.8±1.4 | 43.3±2.8 |
| F-$\mathcal{C}$ | 98.3±0.3 | **100**±0.0 | **100**±0.0 | 98.3±0.2 |
| R-6 | 81.7±3.5 | 77.5±1.7 | 95±1.2 | 93.3±1.5 |
| T-6 | **98.4**±1.3 | **100**±0.0 | **100**±0.0 | **100**±0.0 |

Table 7: Classification accuracy (%) on real-world datasets using GCN, GAT, and GraphSAGE architectures. Results are reported for our target-aware feature ranking method (T-6) and its universal variant (U) which is jointly trained on graphs from multiple domains. Values marked with ✓ indicate the highest accuracy for each dataset and architecture.

|    | MUTAG | MSRC9 | PTCMM | PTCFR | BZR | COX2 | DHFR | Proteins | NCI1 |
|----|-------|-------|-------|-------|-----|------|------|----------|------|
| GCN | | | | | | | | | |
| TA | 90.5✓ | 96.3✓ | 65.0 | 67.5 | 86.9 | 82.2✓ | 85.9✓ | 80.0✓ | 68.0✓ |
| U | 85.7 | 89.7 | 73.3✓ | 70.0✓ | 92.5✓ | 80.0 | 76.2 | 65.0 | 63.1 |
| GAT | | | | | | | | | |
| TA | 88.9✓ | 96.2✓ | 67.6✓ | 70.0✓ | 89.4✓ | 84.5✓ | 75.3✓ | 74.5✓ | 64.2✓ |
| U | 71.4 | 88.5 | 61.7 | 55.0 | 75.0 | 72.2 | 60.0 | 65.2 | 59.7 |
| GraphSAGE | | | | | | | | | |
| TA | 90.4✓ | 95.0✓ | 64.1✓ | 65.0✓ | 95.0✓ | 85.5✓ | 79.6 | 74.0✓ | 75.1✓ |
| U | 83.3 | 89.7 | 57.7 | 62.5 | 82.5 | 78.7 | 83.7✓ | 67.0 | 68.2 |

