# OpenReview forum: "Learning to Rank Features to Enhance Graph Neural Networks for Graph Classification"
_TMLR — Accepted by TMLR_

### Review · Reviewer_CQbo · 2025-08-17

**Summary Of Contributions:**

This paper introduces a novel two-step framework for graph classification that learns to rank and select a compact subset of informative features from a large candidate pool. By integrating feature ranking with GNN training, the method improves accuracy, reduces overfitting, and achieves state-of-the-art results on multiple benchmarks.

**Additional Comments:**

None

**Audience:**

Yes

**Audience Explanation:**

The research topic of graph neural networks for graph classification is very relevant to the interests of TMLR's audience.

**Broader Impact Concerns:**

Since the proposed framework focuses on automated feature selection for graph classification, potential ethical implications include misuse in sensitive domains such as biomedical research, social networks, or financial systems, where biased or misleading feature rankings could amplify existing inequities. A more thorough Broader Impact Statement should discuss these risks, outline safeguards, and clarify the societal contexts in which the method is appropriate or potentially harmful.

**Claims And Evidence:**

No

**Claims Explanation:**

The baseline is too few and outdated

**Requested Changes:**

- My major concern is the lack of a clearly articulated research motivation. It remains unclear what precise role feature selection plays in graph classification and why it is particularly meaningful. The authors do not convincingly differentiate their notion of feature selection from the implicit feature extraction already performed by GNNs, nor do they explain why existing approaches are fundamentally insufficient. These issues require deeper theoretical justification and concrete real-world examples to demonstrate the practical necessity and novelty of the proposed framework.

- It is not evident whether the proposed two-step framework can be easily extended to very large graphs, since the feature ranking step may become computationally demanding. The paper lacks a formal theoretical analysis of time complexity and provides only limited runtime comparisons. Moreover, no systematic scalability experiments are presented to demonstrate how the method performs as graph size or dataset scale increases.

- The experimental evaluation is limited by the choice of baselines, which are relatively few and somewhat outdated. The three main comparison methods were published in 2019, 2021, and 2022, meaning all are at least three years old. To strengthen the credibility of the results, the paper should incorporate more recent baselines from 2024 and 2025, ensuring a fairer and more convincing assessment of the proposed method’s advantages.


- The discussion of related work is not sufficiently clear or comprehensive. In particular, important directions such as graph classification methods based on substructure assembly are missing [1]. The related work section would benefit from a more systematic organization by categories of approaches, followed by a clearer comparison highlighting how the proposed method differs from and improves upon these alternatives.

Refs:

[1] Graph substructure assembling network with soft sequence and context attention[J]. IEEE Transactions on Knowledge and Data Engineering, 2022, 35(5): 4894-4907.

---

> ### Author Response · Authors · 2025-10-09
>
> We thank the Reviewer for their thorough reading and valuable recommendations. Below, we address all comments raised by the Reviewer and indicate where they have been addressed in the revised manuscript. For the Reviewer's convenience, the corresponding changes are highlighted in blue and annotated in the margin with the relevant Reviewer ID.
>
> - *My major concern is the lack of a clearly articulated research motivation. It remains unclear ...* \
> **(Addressed)**  We thank the reviewer for this comment. In the revised introduction (Section 1), we now clearly articulate the motivation and the specific role of feature selection in graph classification, as well as the limitations of existing methods. In particular, we
>     - distinguish between implicit feature extraction performed by GNNs and explicit feature selection, explaining why the latter is necessary (see par. 2 of Section 1),
>     - clarify existing methodological gaps: static feature sets limit generalization, node‑level methods overlook global features, and current feature‑ranking approaches address only node classification (see par. 3 of Section 1), and
>     - motivate the need for data‑driven selection of both local and global features within a unified framework (see par. 4 of Section 1).
>
> - *It is not evident whether the proposed two-step framework can be easily extended to...* \
> **(Addressed)**
> We appreciate the Reviewer’s concern regarding scalability. In our framework, the feature ranking step is *not* the computational bottleneck. Instead, the dominant cost arises from feature computation, similar to other feature-based approaches. This step is performed twice: once using the full candidate feature set on a small subset of the training graphs, and once using the selected subset of features on the remaining graphs. Its complexity is governed primarily by the cost of computing the features themselves.
> Our method offers a significant advantage by avoiding the need to compute all features for all graphs, reducing overall runtime substantially in this way. \
> Following the Reviewer’s recommendation, we have included scalability experiments on substantially larger real-world benchmark graph datasets (see the highlighted text on p. 7 and the last four rows of Table 2 in Appendix B). The corresponding total runtime results are shown in Fig. 5 (note the logarithmic scale on the $y$-axis). While the all-feature baseline required more than 557 hours to process the largest dataset, our approach completed the task in approximately 57 hours, confirming the claimed efficiency (see the highlighted paragraph on scalability on p. 14). \
> Additionally, we now provide a formal runtime analysis (see paragraph ``Runtime Analysis'' in Sect. 4.3.1). Specifically, we derive a condition under which our approach guarantees superior runtime compared to the all-feature baseline, and we demonstrate experimentally that this condition is satisfied by all complex datasets in our study, precisely where the baseline’s runtime becomes critical.
>
> - *The experimental evaluation is limited by the choice of baselines ...* \
> **(Addressed)**
> During our evaluation, we found a substantial discrepancy between the accuracy results reported in the literature and those reproduced using the authors’ publicly available code. To ensure fairness and reproducibility, we decided to consider only results that were (i) published in peer-reviewed conference or journal papers, and (ii) accompanied by publicly available code that could be executed on the corresponding datasets.
> After an extensive search, we identified one further paper from 2024 that meets these criteria and, together with two other methods already considered in the original submission, were able to cover eight of the 13 benchmark datasets. In the revised manuscript, we now compare our approach against SOTA accuracy results with more and less recent baselines (see the first two paragraphs of Sect. 4.3.1).
>
> - *The discussion of related work is not sufficiently clear or comprehensive...* \
> **(Addressed)** We have included the suggested paper (Yang et al., 2022) and thoroughly revised Section 2 on related work. The section is now organized into five categories: (i) substructure and graphlet-based methods, (ii) feature selection and noise reduction, (iii) dual-network architectures, (iv) graph representation learning and pooling, and (v) GNN-based feature ranking learning. For each category, we highlight how our approach differs from and improves upon existing methods.
>
> - *Since the proposed framework focuses on automated feature selection...* \
> **(Addressed)**
> We have added a Broader Impact Statement (see the statement after Sect. 5).

---

### Review · Reviewer_Mk5k · 2025-09-09

**Summary Of Contributions:**

The idea put forth in this paper is learning to rank features that are useful for graph classification when the latter is accomplished via graph neural networks. To achieve this, the authors propose a two-step method that automatically selects a subset of features from a pool of candidate features to improve classification accuracy. In the first step, the importance of each feature for a given graph in a reduced training set is estimated via graph neural networks. In the second step, the suggested algorithm generates feature rankings for the training graphs, and said feature rankings are then aggregated into a global ranking. A top-K subset is selected from this global ranking and used to train a downstream graph classification graph neural network of choice. Experiments on real-world and synthetic datasets show that the proposed algorithm outperforms some baselines tested by the authors, including models using all candidate features.

**Additional Comments:**

-) The paper by Alkhoury et al. reduces the overall novelty.
-) This really is an experiments-oriented paper as the overall algorithm is a marginal extension of current literature.

**Audience:**

Yes

**Audience Explanation:**

The topic is relevant to the community following TMLR.

**Broader Impact Concerns:**

The topic considered is highly important.

**Claims And Evidence:**

Yes

**Claims Explanation:**

While the authors substantiate their main claims with regard to the baselines, the results are mixed and they acknowledge that.

**Requested Changes:**

-) Release source code.
-) List wall-clock times for the entire model — how much overhead the first phase adds overall? Just higher accuracy is not enough if your model is much slower to train.
-) The discussion and justification around the graphlet motifs and their choice/importance is neither well-motivated nor clear.
-) Circular heatmap diagrams are hard to comprehend. Please switch to a different modality of visualization.

---

> ### Author Response · Authors · 2025-10-09
> **Response**
>
> We thank the Reviewer for their thorough reading and valuable recommendations. Below, we address all comments raised by the Reviewer and indicate where they have been addressed in the revised manuscript. For the Reviewer's convenience, the corresponding changes are highlighted in blue and annotated in the margin with the relevant Reviewer ID.
>
> - *Release source code* \
> **(Addressed)** We provide the link to the source code as a footnote on page 4 (the same link was already included in the original submission).
> - *List wall-clock times for the entire model — how much overhead the first phase adds overall? Just higher accuracy is not enough if your model is much slower to train.* \
> **(Addressed)** In the revised manuscript, we now report total wall-clock times (see Fig. 5) and analyze the relationship between accuracy and runtime (see the highlighted paragraph on ``Empirical Runtime and Accuracy'' in Section 4.3.1 and Fig. 6). Our results demonstrate that, on large-scale graphs, our method consistently delivers higher accuracy while significantly reducing overall runtime relative to the all-feature baseline.
> - *The discussion and justification around the graphlet motifs and their choice/importance is neither well-motivated nor clear.* \
> **(Addressed)** We thank the Reviewer for this comment. In response, we have extended one paragraph and added a new one discussing and justifying the use of the graphlet motifs (see the highlighted text on p. 8). The use of graphlets enables us to define a hierarchy of graph classes via logical formulas of increasing structural complexity, all grounded in the same four primitive motifs. This provides a controlled testbed for systematically evaluating the sensitivity and expressiveness of our feature selection mechanism. Our use of graphlets, and the selection of the four particular motifs, is therefore not based on any specific heuristic appeal (e.g., “interesting” or “informative” substructures), but rather on their status as well-defined structural primitives within the candidate feature set that link discrete graph topology and formal logic. This connection offers a rigorous framework for analyzing the capabilities and limitations of our feature selection method.
> - *Circular heatmap diagrams are hard to comprehend. Please switch to a different modality of visualization.* \
> **(Addressed)** We have replaced the circular heatmap diagrams with rectangular grouped bar charts (see Figures 2, 3, and 9).
> - *The paper by Alkhoury et al. reduces the overall novelty. -) This really is an experiments-oriented paper as the overall algorithm is a marginal extension of current literature.* \
> **(Answer)** We thank the Reviewer for this observation and agree that our work builds upon the paradigm introduced by Alkhoury et al.; we explicitly acknowledge their framework as a foundation in the manuscript. While their paper focuses on node classification, our work addresses graph classification, a setting that introduces distinct challenges. The adaptation of their framework to graph classification is not immediate; it requires the development of new techniques tailored to this context. These techniques are not straightforward and involve careful consideration of new challenges that do not arise in the node classification scenario. \
> Moreover, as our work is experiment-oriented, we place special emphasis on evaluating the effectiveness and limitations of these adaptations through systematic empirical analysis, offering insights that may be of interest to both theory and practice. We believe that by extending and rigorously testing the original paradigm in a new problem class, our work constitutes a meaningful and non-trivial contribution to research on GNNs.

---

> > ### Comment · Reviewer_Mk5k · 2025-11-08
> > **Response**
> >
> > The authors addressed all of my comments. I appreciate the new experiments as well as the responses to the rest of the reviewers.

---

### Review · Reviewer_a3BA · 2025-09-26

**Summary Of Contributions:**

The goal of this paper is to improve graph classification performance in GNNs by automatically learning and selecting a small, informative subset of node- and graph-level features from a large candidate pool. The paper introduces a two-step method that enhances graph classification by ranking and selecting features to augment GNN inputs. In Step 1, a small subset of training graphs is sampled, and all 124 candidate features—spanning graphlets, aggregated node statistics, and global graph properties—are computed. A random forest classifier is trained per graph using comparisons with other graphs of different labels to produce feature importance rankings. These local rankings are aggregated and used to train a Feature-Ranking GNN (FR-GNN), which generalizes feature ranking predictions to unseen graphs. In Step 2, the FR-GNN is applied to the remaining training graphs to produce a global feature ranking. The top K features from this ranking are then computed and used to augment the training data for a Graph Classification GNN (GC-GNN). This GNN is trained using the compact feature set. The method is benchmarked against baselines that use no features, all features, or randomly selected features, across real-world and synthetic datasets using three GNN architectures (GCN, GAT, GraphSAGE). Evaluation also includes runtime comparisons and tests of statistical significance. Across nine real-world and multiple synthetic datasets, the proposed method consistently outperforms baselines, achieving up to 10% higher accuracy than using all features and showing statistical significance. Even with only six selected features, it matches or exceeds state-of-the-art performance, while reducing overfitting and computation time on large graphs.

**Audience:**

Yes

**Audience Explanation:**

Although learning to rank is a mature area of research, there is still room for improvement of current techniques. Its combined application on graph neural networks for graph classification will be of interest to most researchers and readers in the TMLR audience.

**Broader Impact Concerns:**

This is a methods paper for ranking. So a discussion on what implications the technique may have on datasets of associated to human subjects could be important to at least discuss.

**Claims And Evidence:**

Yes

**Claims Explanation:**

The paper presents an effective feature reduction method that selects highly informative features to improve classification while reducing overfitting and runtime. Its main strength is empirical, as it provides extensive evaluations of the proposed ideas, which are tested across diverse domains (e.g., chemistry, bioinformatics, computer vision) and synthetic settings with known ground truths. The performance improvements are validated using the Kruskal-Wallis test and Dunn’s post-hoc test, confirming significance over the baselines.
However, the paper also has room for improvement. First, there is no formal proof of optimality for the selected feature subsets or convergence guarantees for the method. Second, the FR-GNN is trained separately for each dataset. Thus, attempts to build a global FR-GNN yield lower performance gains. Finally, the selection of the number of top features (e.g., K=6) is empirically driven; no adaptive mechanism for optimal K per dataset is proposed.

**Requested Changes:**

Some elements of the paper can be further clarified. For instance, what conditions must be met for the feature selection to guarantee near-optimal classification performance? Can bounds on generalization error or subset optimality be derived?
Furthermore, can we design a data-driven strategy to dynamically choose the number of top features K, possibly via validation accuracy curves or information-theoretic criteria? For instance, why not apply a Bayesian optimization over K yield better generalization?
Ultimately, some clarity about the applicability would be beneficial. How can the FR-GNN be adapted or fine-tuned to generalize feature rankings across diverse datasets while preserving task-specific accuracy? Would meta-learning or transfer learning approaches help train a universal FR-GNN?

---

> ### Author Response · Authors · 2025-10-09
>
> We thank the Reviewer for their thorough reading and valuable recommendations. Below, we address all comments raised by the Reviewer and indicate where they have been addressed in the revised manuscript. For the Reviewer's convenience, the corresponding changes are highlighted in blue and annotated in the margin with the relevant Reviewer ID.
>
> - *There is no formal proof of optimality for the selected feature subsets or convergence guarantees for the method. What conditions must be met for the feature selection to guarantee near-optimal classification performance? Can bounds on generalization error or subset optimality be derived?* \
> **(Addressed)** We thank the Reviewer for raising this important point. Regarding optimality, we note that selecting a feature set of smallest cardinality that is optimal on the training data is NP-hard; this can be shown by a polynomial reduction e.g. from the set covering problem. Exhaustive search over all possible subsets is computationally infeasible (in our setting we consider 124 candidate features). While we currently do not have answers to the Reviewer’s two theoretical questions, we fully recognize their importance. The first one has already been formulated in the original manuscript; we explicitly state the second one as an open question in the revised manuscript (see  the highlighted text in Sect. 5).
> - *How can the FR-GNN be adapted or fine-tuned to generalize feature rankings across diverse datasets while preserving task-specific accuracy? Would meta-learning or transfer learning approaches help train a universal FR-GNN?* \
> **(Answer)** We thank the reviewer for this comment. In a prior study, we implemented a universal FR-GNN designed to generalize feature rankings across datasets through intra- and inter-dataset graph sampling. The model jointly trained on graphs from multiple domains to learn transferable feature importance patterns. \
> However, as summarized in Table 7, the target-aware FR-GNN (our method) consistently outperformed the universal variant, achieving the highest accuracy in 23 out of 27 runs. This result indicates that feature relevance is highly dataset-dependent, and while universal ranking captures broad trends, it sacrifices task-specific discriminative power.
> - *Can we design a data-driven strategy to dynamically choose the number of top features K, possibly via validation accuracy curves or information-theoretic criteria? For instance, why not apply a Bayesian optimization over K yield better generalization?* \
> **(Addressed)**
> We thank the Reviewer for this comment. We agree that dynamically selecting $K$ using a validation set constitutes a more principled approach. Our framework is compatible with such strategies, including Bayesian optimization, provided a sufficiently small validation set is available. In particular, any dynamic selection of $K$ can be integrated into our pipeline, as long as retraining the GC-GNN model, the only step that must be repeated from scratch, is computationally feasible. We have clarified this point in the revised manuscript (see the highlighted paragraph on "Choice of $K$" on p. 12).

---

### Decision · Action_Editor_P8pg · 2025-11-10

**Recommendation:** Accept as is

**Audience:**

Yes

**Audience Explanation:**

Explicit feature ranking to improve graph classification is clearly a relevant relevant to TMLR readers working on graph learning and representation learning.

**Claims And Evidence:**

Yes

**Claims Explanation:**

The three reviewers are positive about this paper. After the rebuttal phase all of them were satisfied with the authors' responses.

Although the contribution is incremental (it heavily relies on previous work by the same authors) it is still significant, as it extends the methodology from node-level to graph classification. This extension requires additional considerations, and the work backs this up with extensive experiments showing consistent improvements against baselines.

The paper improved after the revision: the motivation was clarified and the related work section reorganized with added references, the experimental section was strengthened with an updated baseline, and they added runtime and scalability analysis, and improved the overall presentation.